# Towards an improved understanding of the impact of clouds and precipitation on the representation of aerosols over the Boreal

# Forest in GCMs

4

3

- 5 Sini Talvinen<sup>1,2</sup>, Paul Kim<sup>3</sup>, Emanuele Tovazzi<sup>3</sup>, Eemeli Holopainen<sup>1,4,5</sup>, Roxana Cremer<sup>2,\$</sup>, Thomas
- 6 Kühn<sup>6</sup>, Harri Kokkola<sup>1,4</sup>, Zak Kipling<sup>7</sup>, David Neubauer<sup>78</sup>, João C. Teixeira<sup>9</sup>, Alistair Sellar<sup>9</sup>, Duncan
- Watson-Parris<sup>10</sup>, Yang Yang<sup>11</sup>, Jialei Zhu<sup>12</sup>, Srinath Krishnan<sup>13</sup>, Annele Virtanen<sup>1</sup> and Daniel G.
- 8 Partridge<sup>3</sup>

9

- <sup>1</sup>Department of Technical Physics, University of Eastern Finland, Kuopio, 70210, Finland
- <sup>2</sup>Department of Environmental Science and Bolin Centre for Climate Research, Stockholm University, Stockholm, 10691,
- 12 Sweden
- 13 Department of Mathematics and Statistics, University of Exeter, Exeter, EX4 4QF, United Kingdom
- <sup>4</sup>Atmospheric Research Centre of Eastern Finland, Finnish Meteorological Institute, Kuopio, 70211, Finland
- 15 SInstitute for Chemical Engineering Sciences of Foundation for Research and Technology Hellas, Patras, 26504, Greece
- 16 Weather and Climate Change Impact Research, Finnish Meteorological Institute, Helsinki, 00101, Finland
- <sup>7</sup>European Centre for Medium-Range Weather Forecasts, Reading, RG2 9AX, United Kingdom
- 18 Institute for Atmospheric and Climate Science, ETH Zürich, Zurich, Switzerland
- <sup>9</sup>Met Office Hadley Centre, Exeter, EX1 3PB, United Kingdom
- 20 <sup>10</sup>Scripps Institution of Oceanography and Halıcıoğlu Data Science Institute, University of California San Diego, La Jolla,
- 21 CA 92093, United states
- <sup>11</sup>School of Environmental Sciences and Engineering, Nanjing University of Information Science and Technology,
- 23 Nanjing, Jiangsu, China
- <sup>12</sup>Institute of Surface-Earth System Science, School of Earth System Science, Tianjin University, Tianjin, 300072, China
- <sup>13</sup>CICERO Center for International Climate Research, Oslo, 0349, Norway
- <sup>\$</sup>now at Leibniz Institute for Tropospheric Research, Leipzig, 04318, Germany

- Correspondence to: sini.talvinen@uef.fi, D.G.Partridge@exeter.ac.uk
- Abstract
- Global climate models (GCMs) face uncertainties in estimating Earth's radiative budget due to aerosol-cloud interactions
- (ACI). Accurate particle number size distributions (PNSDs) are crucial for improving ACI representation, requiring
- precise modelling of aerosol sources and sinks. Using a Lagrangian trajectory framework, we examine how clouds and
- precipitation influence aerosols during transport, and thereby influence aerosol-cloud relationships in the boreal forest.
- Two GCMs, the United Kingdom Earth System Model (UKESM1) and ECHAM6.3-HAM2.3-MOZ1.0 with the
- SALSA2.0 aerosol module (ECHAM-SALSA), are complemented with model-derived trajectories and evaluated against
- in-situ observations, which are accompanied by reanalysis trajectories. Overall aerosol-precipitation trends are similar
- between GCMs and observations. However, seasonal differences emerge: in summer, UKESM1 exhibits more efficient
- aerosol removal via precipitation than ECHAM-SALSA and observations, whereas in winter, the opposite is observed.
- These differences coincide with key variables controlling aerosol activation, such as sub-grid scale updraught velocities
- and PNSDs. For example, in winter, removal of total aerosol mass in ECHAM-SALSA was stronger than in UKESM1,
- coinciding with higher activated fractions and larger sub-grid scale updraught velocities in ECHAM-SALSA. For both
- GCMs, cloud processing along trajectories increased SO<sub>4</sub> mass, mainly in the accumulation mode, consistent with
- do observations and model parametrizations. Discrepancies arise more from differences in PNSDs and updraught velocities
- than from wet removal parametrizations, an example being the underrepresentation of small particles in UKESM1. While

- our findings are representative of boreal region with predominantly stratiform precipitation, further work is needed to
- evaluate their applicability to other regions.

#### 1 Introduction

61

63

68

8182

Atmospheric aerosol particle concentrations are influenced by their sources and sinks which affect their lifetimes in the atmosphere, and also play a significant role in our climate system through different mechanisms. One of the most important mechanisms are aerosol-cloud interactions (ACI), which are still causing the largest uncertainties on the effects of aerosols on Earth's radiative budget in global climate models (GCMs, Boucher, 2013; Watson-Parris et al., 2019; Bellouin et al., 2020; Forster et al., 2021), and therefore partly masking the warming effect by greenhouse gases (Bauer et al., 2022; Quaas et al., 2022). It is critical, therefore, that the microphysical processes influencing ACIs are well understood and accurately modelled. To accurately simulate ACI in GCMs, the aerosol number size distributions need to be correctly described (e.g., Mann et al., 2010). Traditionally, discrepancies in particle size distributions between observations and models exceed those between modal and sectional approaches, with sectional methods dividing the distribution into discrete size bins (Mann et al., 2012). However, larger differences in concentrations may emerge when chemistry of the aerosols is inspected (Laakso et al., 2022). On the other hand, to accurately represent the aerosol number size distributions, GCMs also need to accurately represent the source and sink processes that act on the aerosol during its lifetime and transport in the atmosphere. The impact of precipitation on the evolution of the size distribution is very important (e.g., Browse et al., 2014; Khadir et al., 2023), but remains a major uncertainty in the GCMs. Often, when GCM parametrizations are assessed the models are evaluated against observations or other GCMs by inspecting differences in averages of variables (or relationships between multiple variables) over certain time spans (e.g., Blichner et al., 2024; Gliß et al., 2021; Labe and Barnes, 2022; Maher et al., 2021; Pathak et al., 2023) in a Eulerian perspective. However, GCM evaluations in which the evolution of aerosols and other variables is followed over both time and space in more detail using GCM Lagrangian trajectory-based evaluation frameworks that have been recently introduced (e.g., Kim et al., 2020). Such frameworks pave the way for the development of more rigorous observational constraints on uncertain physical and chemical aerosol processes for GCM evaluation, by including temporal and spatial information associated with the air-mass history. ACIs include scavenging of aerosol particles by precipitation, cloud droplets and ice crystals. Wet scavenging is one of the most efficient removal routes of particles from the atmosphere (e.g., Ohata et al., 2016; Liu et al., 2020). Wet scavenging of aerosol particles can be further divided into in-cloud scavenging and below cloud scavenging. Wet scavenging via in-cloud scavenging involves the loss of aerosol particles when they become activated into cloud droplets or ice crystals (nucleation scavenging) which can then further collide with interstitial aerosols in-cloud (e.g., Ohata et al., 2016; Seinfeld and Pandis, 2016). Below-cloud scavenging concerns the removal of aerosol by rainfall from the collection of particles due to collisions with falling raindrops and snow and ice from precipitation (e.g., Ohata et al., 2016). Current understanding identifies the contribution of in-cloud scavenging, followed by removal via precipitation to be, on average, the most important sink globally for accumulation mode particles (particle diameter  $d_p \sim 100-1000$  nm). Ultrafine ( $d_p <$ 100 nm) and coarse particles (d<sub>p</sub> > 1 μm), on the other hand, are more efficiently removed by below-cloud scavenging (e.g., Andronache, 2003; Textor et al., 2006; Croft et al., 2009; Ohata et al., 2016). In addition to wet scavenging, clouds can also alter the particle properties through aqueous phase oxidation processes. For example, sulfate production due to

oxidation of gaseous sulfur dioxide inside clouds is considered as one of the most important mass addition processes for

sulfate (e.g., Ervens, 2015 and references therein). Production of organics through aqueous phase processes has also been

reported in some environments (e.g., Ervens et al., 2018; Lamkaddam et al., 2021).

Investigation of the effects of precipitation and clouds has traditionally been Eulerian, in which local estimates of precipitation are employed (e.g., Wang et al., 2021). Lagrangian approaches, in which air mass trajectories are exploited to examine the effects of precipitation on aerosols and their composition as the air masses travel to the receptor location, have, however, increased in popularity during the recent years (Dadashazar et al., 2021; Heslin-Rees et al., 2024; Isokääntä et al., 2022; Kesti et al., 2020; Khadir et al., 2023; Tunved et al., 2004, 2013; Tunved and Ström, 2019). These types of studies can provide significantly more detailed insights by considering the interplay between aerosols, clouds and precipitation during air mass history, that cannot be achieved using Eulerian approaches. All these studies investigated how the total accumulated precipitation experienced along air-mass trajectories derived from reanalysis data affects a particle size distribution measured at a specific receptor site. Tunved et al. (2013), for example, investigated aerosols in the Arctic (Zeppelin station, Ny-Ålesund, Norway) and observed strong removal of sub-micron particulate mass up to 10 mm of accumulated precipitation. They suggested the in-cloud scavenging (followed by removal via precipitation) is the dominant removal pathway, as larger particles showed first a decrease in their concentration as a function of accumulated precipitation during transport, followed by the removal smaller sizes. Kesti et al. (2020) studied aerosols at the humid tropical monsoon climate in the Maldives, and observed more efficient removal of the accumulation mode particles with increasing accumulated precipitation, when compared to the smaller particle sizes. Dadashazar et al. (2021) studied subtropical environments in Bermuda and concluded that PM<sub>2.5</sub> mass experienced the strongest sensitivity to accumulated precipitation up to 5 mm whereas precipitation exceeding this limit had no major effects on the particulate mass. Khadir et al. (2023) further reported that precipitation can, in some instances, serve as a source of aerosols.

In addition to the effects of precipitation for aerosols, a previous study by Isokääntä et al. (2022) used relative humidity (>94%) as a proxy for in-cloud exposure in boreal air masses and found a pronounced increase in sulfate mass in air masses recently influenced by non-precipitating clouds, while no significant aqueous-phase production of organic aerosol was observed—likely due to dominant gas-phase biogenic sources. This is consistent with findings from central Sweden (Graham et al., 2020). These earlier results suggest that sulfate may be more strongly affected by cloud processing and wet removal than organic aerosol, with removal efficiency likely influenced by factors such as precipitation timing, aerosol type, and the stage of the air mass trajectory. Our study builds on this by exploring these aspects across multiple models and observations, employing the GCM Lagrangian evaluation framework presented by Kim et al. (2020). With this framework air mass trajectories can be obtained from global GCM simulations. This is achieved by co-locating multiple variables (for example, aerosol size distribution and chemical composition) from the GCMs to air mass trajectories calculated from the GCM meteorological data (Kim et al., 2020). This methodology allows us to transparently evaluate and compare the wet scavenging and aqueous-phase processing between the observations and GCMs within the Lagrangian trajectory framework in unprecedented detail.

This study compares the effects of wet processing (wet removal and aqueous-phase processing) on modelled aerosol size distributions with long-term observations from Hyytiälä, Finland. Observational trajectories are based on ERA-Interim reanalysis, while model trajectories are calculated using meteorology data from GCM AMIP-style simulations in which wind fields were nudged to ERA-Interim. The GCMs used in this study include UKESM1 (United Kingdom Earth System Model, e.g., Sellar et al., 2019) and ECHAM6.3-HAM2.3-MOZ1.0 with sectional aerosol module SALSA2.0 (hereafter ECHAM-SALSA, Stevens et al., 2013; Kokkola et al., 2018; Tegen et al., 2019). Both GCMs are part of the Aerosol Comparisons between Observations and Models (AeroCom) Phase III GCM Trajectory Experiment (GCMTraj) in which a comparison between the GCMs against reanalysis meteorology was conducted for the years between 2009 and 2013. In this study the simulations for UKESM1 and ECHAM-SALSA cover the years from 2005 to 2018 which are also available

from the observations. Comparison between modal (UKESM1) and sectional (ECHAM-SALSA) approaches for estimating the aerosol microphysics provides additional insight into the model behaviour via this Lagrangian evaluation approach. The Hybrid Single-Particle Lagrangian Integrated Trajectory model (HYSPLIT; Draxler and Hess, 1998; Stein et al., 2015) is employed to obtain the backward air mass trajectories. A key difference between our study and previous work, including Isokääntä et al. (2022), is our focus on stratiform precipitation rather than total precipitation. Stratiform precipitation is the dominant type in mid- and high-latitude regions (30–60° from the equator and poleward), whereas tropical regions are typically influenced by convective systems (e.g., Schumacher and Funk, 2023). Since our study area is primarily the boreal forest region of northern Europe, stratiform precipitation is most relevant. The differing impacts of precipitation types on aerosols have also been highlighted by Khadir et al. (2023), who showed that recent tropical precipitation—largely convective—can be linked to downdrafts that transport small particles from higher altitudes to the boundary layer (see also Franco et al., 2022; Machado et al., 2021; McCoy et al., 2021; Williamson et al., 2019).

The aim of our research can be summarized into two main objectives (1-2):

- 1. Do the relationships between aerosols and experienced precipitation during transport differ between the measurements and GCMs and what are the drivers for the observed differences?
- 2. Do the GCMs exhibit similar increase in sulfate mass due to in-cloud production as the observations and are the observed effects reasonable when compared to model parametrizations?

We start our investigation in Sect. 2 by first introducing the observational datasets, followed by summarising the GCM simulations along with details on the air mass trajectory calculations and data co-locations employed in this work. The aerosol properties at the measurement station (Hyytiälä, Finland) are given in Sect. 3 as a necessary background for the following Lagrangian analysis. The relationships between precipitation and aerosol mass and number in the Lagrangian framework are presented first (Sect. 4.1-4.3), followed by a process-chain type evaluation (Sect. 4.4) to understand the driving forces in the relationships. Finally, in Sect. 5, the effects of aqueous-phase processing are presented, followed by overall conclusions (Sect. 6) and outlook (Sect. 7).

#### 2 Data and methods

160

161

#### 2.1 Observations at SMEAR II

Observational data used in this study include long-term measurements of aerosol number size distributions and particle chemistry from SMEAR II (Station for Measuring Ecosystem-Atmosphere Relations in; Hari and Kulmala, 2005) and are 151 described in detail in Isokääntä et al. (2022) and the references therein. SMEAR II station (Hyytiälä, Finland) is classified 152 153 as a rural environment, surrounded by relatively homogenous Scots pine (Pinus sylvesteris) forest. In this work particle 154 number size measurements (covering particle diameters between 3-1000 nm) obtained with a differential mobility particle sizer (DMPS, e.g., Aalto et al., 2001) are utilized. Chemical composition (organics, sulfate, and equivalent black carbon) 155 156 of the particles in the sub-micron range were derived from an aethalometer (e.g., Drinovec et al., 2015) and aerosol 157 chemical speciation monitor (ACSM, Ng et al., 2011). The dataset for particle number size measurements spans 2005-158 2018, slightly shorter than in Isokääntä et al. (2022), to match the GCM simulation period. The ASCM data extends from 2012 to 2018. 159

#### 2.2 Summaries of the GCMs used in this study

#### 2.2.1 UKESM1

- The United Kingdom Earth System Model (UKESM1) configuration used in this study uses the atmospheric and land
- components following the protocol set by the Atmospheric Model Intercomparison Project (AMIP, Eyring et al., 2016).
- The atmospheric component of the model is based on the Global Atmosphere 7.1 (GA7.1) and the Global Land 7.0
- (GL7.0) configurations, as described by Walters et al. (2019). These are part of the Hadley Centre Global Environment
- Model version 3 (HadGEM3; Hewitt et al., 2011), which is coupled to the terrestrial carbon/nitrogen cycles (Sellar et al.,
- 2019). It includes interactive stratosphere–troposphere chemistry from the from the UK Chemistry and Aerosol (UKCA)
- model (Archibald et al., 2020; Morgenstern et al., 2009; O'Connor et al., 2014).
- Following the AMIP protocol, sea surface temperature and sea ice are taken from the unmodified dataset of Durack et al.
- (2017) and horizontally interpolated to the model resolution. In this setup, the dynamic vegetation model (Cox, 2001) is
- turned off. Instead, prescribed vegetation from a historical coupled UKESM1 simulation is used to maintain consistent
- land-use forcing between the coupled and AMIP experiments. In a similar fashion, seawater concentrations of dimethyl
- sulfide (DMS) and chlorophyll-a monthly climatologies are taken from the coupled historical experiment and are used by
- the atmosphere model top calculates fluxes of DMS and primary marine organic aerosol (Mulcahy et al., 2020).
- The simulations were nudged to ERA-Interim reanalysis (Dee et al., 2011; Telford et al., 2008) u/v (horizontal and
- vertical), wind fields and surface pressure following the setup design for the AeroCom GCMTraj phase III experiment.
- The model resolution for these configurations was 1.875° × 1.25° longitude–latitude, corresponding to a horizontal
- resolution of ~135 km in the midlatitudes. The model has 85 vertical levels which are divided such that 50 levels are
- between 0 and 18 km and the remaining 35 levels cover heights between 18 and 85 km.
- Atmospheric composition within UKESM1 is implemented as part of the UKCA model. Within UKCA, the Global Model
- of Aerosol Processes (GLOMAP; Mann et al., 2010; Mulcahy et al., 2020) is used. This scheme simulates multicomponent
- global aerosols, including, for example, sulfate, black carbon, and organic matter. The aerosol particle size distribution is
- represented using five log-normal modes, nucleation soluble, Aitken soluble, accumulation soluble, coarse soluble and
- Aitken insoluble visualized in Figure S1. More details, including the size ranges for each aerosol mode, are presented in

Sect. S1.1. The GLOMAP model also includes various microphysical processes that affect the evolution of aerosol properties. Wet scavenging processes in UKESM1, including below-cloud (impaction), in-cloud (nucleation) and plume scavenging are summarized in Sect. S2 and references therein. As a key difference to ECHAM-SALSA (Sect. 2.2.2) concerning the aerosol parametrizations, new particle formation in the boundary layer is not implemented in this version of UKESM1 (Mulcahy et al., 2020).

For this study the AeroCom GCMTraj UKESM1 simulations (2009-2013) were extended for the period from 2005 to 2018 to facilitate robust statistical comparison with the aerosol size distributions and composition measurements obtained from SMEAR II. The model output fields were extracted at high temporal resolution (3-hourly output) for all model levels (when available, otherwise noted as surface). The diagnostics fields utilized in this work (see also Table S4) are aerosol particle size distribution variables (number concentrations and dry diameters for each aerosol mode), chemical components including mass mixing ratios of sulfate noted here as SO<sub>4</sub> (extracted as sulfuric acid H<sub>2</sub>SO<sub>4</sub> and then converted, see Sect. S1.1), organic matter (noted here as OA) and black carbon (BC), total (including both liquid rain and snow) stratiform and convective precipitation at the surface, dry air density, sub-grid scale updraught velocity, number of activated particles, total precipitation at the surface, relative humidity and cloud fractions. Additionally, from UKESM1, wet scavenging coefficients (representing removal within the whole atmospheric column) for the different removal processes (nucleation, impaction and plume) and species (OA, H<sub>2</sub>SO<sub>4</sub> and BC), SO<sub>2</sub> concentrations, and both vertically resolved and surface liquid stratiform precipitation are inspected. These variables and/or variables derived from them are co-located to the UKESM1 derived HYSPLIT back-trajectories as described in Sect. 2.3.

#### 2.2.2 ECHAM-SALSA

ECHAM6.3-HAM2.3-MOZ1.0 is a global aerosol-chemistry-climate model consisting of the atmospheric general circulation model ECHAM (Stevens et al., 2013) coupled with the Hamburg Aerosol Model HAM (Tegen et al., 2019) and chemistry model MOZ (Schultz et al., 2018). For this work, as for UKESM1, simulations follow AMIP style runs following the AeroCom phase III GCMTraj experiment setup. Therefore, as for UKESM1, the u/v wind fields and surface pressure were nudged towards ERA-Interim reanalysis data. In addition, the sea surface temperature and sea ice cover were prescribed based on monthly mean climatologies obtained from the AMIP project (Eyring et al., 2016). The model solves atmospheric circulation with vertical gridding of 47 layers extending roughly up to 80 km. Model horizontal resolution for these configurations is  $1.875^{\circ} \times 1.875^{\circ}$  longitude–latitude.

ECHAM6.3-HAM2.3-MOZ1.0 is paired with the sectional aerosol microphysics model SALSA2.0 (ECHAM-SALSA) in which the size distribution is divided into 3 subranges ( $d_{p1} = 3 - 50$  nm,  $d_{p2} = 50 - 700$  nm and  $d_{p3} = 700$  nm - 10  $\mu$ m)

including 10 size classes in logarithmical size space. Subranges  $d_{p2}$  and  $d_{p3}$  include parallel size classes for insoluble and

soluble aerosol species, making the total number of size classes 17 (Kokkola et al., 2018), visualized in Figure S1. More

details of the subranges and their compositions are given in Sect. S1.2. Additional details of the aerosol processes

calculated in SALSA2.0 can be found in Kokkola et al. (2018) and Holopainen et al. (2020). Wet scavenging

parametrizations are summarized in Sect. S2 for below- and in-cloud scavenging.

As for UKESM1, simulations cover the years from 2005 to 2018 for ECHAM-SALSA. Data output is also 3-hourly and vertically resolved unless the variable is noted as surface variable. The diagnostics extracted from ECHAM-SALSA (see also Table S4) include aerosol particle size distribution variables (number concentrations and dry diameters for each size class), chemical components including mass mixing ratios of sulfate (SO<sub>4</sub>), organics (noted here as OA) and black carbon

- (BC), total (including both liquid rain and snow) stratiform and convective precipitation at the surface, dry air density,
- sub-grid scale updraught velocity, number of activated particles, total precipitation at the surface, relative humidity and
- cloud fractions. Similar to UKESM1, these variables and/or variables calculated from them are co-located to the ECHAM-
- SALSA derived HYSPLIT back-trajectories as described in Sect. 2.3.

# 2.3 Air mass trajectory calculations and data co-location

#### 2.3.1 HYSPLIT

245

- The 4-day (96 h) back trajectories arriving at SMEAR II were calculated by version 5.1.0 of the HYSPLIT (Stein et al.,
- 2015) model for the period from January 2005 to December 2018. The 4-day long back trajectories were used to ensure
- consistency with the results from Isokääntä et al. (2022). In addition, this is typically a long enough period for slowly
- moving air masses to travel to the boreal environment from high arctic and marine areas. Arrival height of the trajectories
- to the receptor station was set to 100 m above the ground level. To obtain the GCM derived trajectories, the meteorological
- fields from the GCMs were first converted into a consistent netCDF4 format which was then converted into the ARL
- packed HYSPLIT4 compatible format (Kim et al., 2020). The GCM and ERA-Interim (Dee et al., 2011) reanalysis
- meteorological datasets required for the HYSPLIT4 trajectory calculations were re-gridded to a consistent 1° horizontal
- resolution. The vertical discretization of the GCM variables was provided on terrain-following hybrid sigma-pressure
- levels. In UKESM1, however, the native output is on hybrid height levels, which is not supported by HYSPLIT.
- Therefore, UKESM1 was output on fixed pressure levels, selected to closely match the ERA-Interim pressure levels.
- Trajectories were calculated for every 3<sup>rd</sup> hour for both reanalysis data and the GCMs, corresponding to GCM output
- resolution. This led to 8 trajectories per day, a total of 40896 air mass trajectories between 2005-2018 before applying
- any pre-processing and temporal harmonization of the data (Sect. 2.4). Hereafter, when discussing observational data
- coupled with the ERA-Interim back-trajectories, those are referred as observations unless mentioned otherwise. It should
- be noted that reanalysis data is not interchangeable with observations but is used as a proxy in this study.

# 2.3.2 Co-location of GCM data along the air mass trajectories

- The variables from the GCMs described in Sect. 2.2.1 and 2.2.2 were temporally (time), spatially (latitude, longitude) and
- vertically (variables which covered different model or pressure levels) co-located to the GCM derived air mass
- trajectories. In short, a co-locator tool (Kim et al., 2020) based off the Community Intercomparison Suite (CIS, Watson-
- Parris et al., 2016) was used to co-locate 4-dimensional data which uses hybrid altitude coordinates. As the default
- interpolator within CIS has often difficulties co-locating to the near-surface trajectory points (due to surrounding grid-
- boxes being at the boundaries of the data domain), the modified co-locator provided more flexibility for the interpolation
- of these near-surface points. This is relevant also in this work, as for our surface site the trajectories can also travel at low
- altitudes. In this improved co-locator, when the linear interpolation in the near-surface trajectories would result into a
- missing value, nearest-neighbour interpolation is used instead. Thus, extrapolation of values can be avoided and
- information for trajectory points that are within the data domain retained. The co-located GCM data from the air mass
- trajectory arrival times, i.e., times when the air mass is located at SMEAR II, are used to represent the conditions at
- SMEAR II, facilitating direct comparison to observational data obtained at the site.
- A difference to Isokääntä et al. (2022) where the ERA-Interim precipitation internally processed by HYSPLIT onto
- trajectories coordinates was used, is that the raw precipitation fields from ERA-Interim are employed in this work by co-

260 locating them to the air mass trajectories in a post-processing step (as for the variables extracted from GCMs described

above). This approach was chosen to retain the original numerical precision of ERA-Interim (and GCM) precipitation

data, ensuring accurate alignment with co-located GCM variables (e.g., aerosol size distributions and chemical

composition), which HYSPLIT does not provide. Here, "consistency" refers to numerical accuracy rather than matching

data sources.

#### 2.4 Data harmonization between measurements and GCMs

#### 2.4.1 Temporal co-location and data pre-processing

- The data from the measurements (1-hourly averages) conducted at SMEAR II was temporally co-located with the ERA-
- Interim derived back-trajectory arrival times (3-hourly). Additionally, the GCM derived trajectories (3-hourly) were only
- co-located with the times when aerosol observations were available. By adopting this approach, only GCM trajectories
- corresponding to existing data points in observations were retained and utilized in further analysis. The importance of
- temporal co-location for model evaluation is discussed, for example, in Schutgens et al. (2016). Harmonisation of the
- measured aerosol size distribution and composition with the corresponding variables available from the GCMs are
- described in Sect. 2.4.2 and 2.4.3.
- For consistency with Isokääntä et al. (2022) identical pre-processing is applied here to the in-situ aerosol observations
- before the temporal co-location described above. Thus, data points for which the measured wind direction was between
- 120 and 140 degrees were removed due to possible influence of strong VOC (volatile organic compound) emissions from
- the local sawmill (Heikkinen et al., 2020; Liao et al., 2011). In addition, trajectories crossing the area of Kola Peninsula
- were excluded as in Isokääntä et al., (2022) due to strong pollution sources within the area (Heikkinen et al., 2020;
- Kulmala et al., 2000; Riuttanen et al., 2013). This led to aerosol size distribution data covering the years between 2005
- and 2018 (number of final data rows/trajectories: 30688) and aerosol chemical composition for the years between 2012
- and 2018 (number of final data rows/trajectories: 6174). Distribution of the data points over the years are shown in Figures
- S2 and S3.

#### 2.4.2 Aerosol particle number size distribution

- The DMPS (differential mobility particle sizer, e.g., Aalto et al., 2001) observations include 51 size bins in the observed
- size range (dp = 3-1000 nm). For UKESM1, complete log-normal particle number size distributions (Seinfeld and Pandis,
- 2016) were calculated by using the modal parameters (dry diameters, number concentrations and geometric mean
- diameters) given by the model. The number size distribution is discretised into the same size grid as the observations i.e.,
- the bin midpoints are identical to the ones available from the DMPS measurements. This approach was possible as in
- SMEAR II the size grid DMPS applies stays constant over the whole investigated period. This harmonization was
- conducted for each hour along the air mass trajectories using the co-location approach described in Sect. 2.3.2 as
- UKESM1 provided all needed modal parameters for calculation of the full particle number size distributions (PNSD)
- along the trajectories.
- For ECHAM-SALSA, the number concentrations of soluble and insoluble bins (i.e., size classes) were added together for
- each size bin. To make the logarithmic number size distribution comparable to UKESM1 data and DMPS measurements,
- the values within each size bin (i) were divided by the logarithm of the maximum size d<sub>i,max</sub> minus the logarithm of the
- minimum size d<sub>i,min</sub> i.e., by log<sub>10</sub>(d<sub>i,max</sub>)-log<sub>10</sub>(d<sub>i,min</sub>) for that size bin (see Table S3). Similar to UKESM1, this was

- conducted along the trajectories. For aerosols, ECHAM-SALSA bins ranging from 3.0 nm to 1700 nm in diameter are studied, as by strictly limiting to sub-micron bins ( $\leq$  700 nm), the largest sub-micron particles (700 nm < d<sub>p</sub>  $\leq$  1000 nm) that do contribute to the total particle mass, would be lost. Sensitivity analysis was conducted including only the sub-micron bins, and none of the conclusions changed.
- Integrated variables, such as total number and mass concentrations (for submicron particles) were calculated from the particle number size distributions by assuming the particles are spherical and have a constant density of  $\rho = 1.6$  g cm<sup>-3</sup>. This density corresponds to the average density of particles observed at SMEAR II (e.g., Häkkinen et al., 2012). Again, these quantities were calculated for each hour (i.e., 96 data points, see Sect. 2.3.1) along every single air mass trajectory.

#### 2.4.3 Chemical composition

308309

316317

Observational data for organic aerosol (hereafter OA) and sulfate (hereafter SO<sub>4</sub>) was obtained using observations from ACSM (aerosol chemical speciation monitor, Ng et al., 2011) which is most efficient at measuring particles with ~ 75-650 nm of vacuum aerodynamic diameter, passing through particles up to 1 µm (Liu et al., 2007). For UKESM1, Aitken and accumulation mode are used in this context by summing the mass mixing ratios (MMR, kg of species per kg of air) of these modes, including both soluble and insoluble modes when available. Due to the definition of the modes in UKESM1, these correspond to particle diameters between 10-500 nm (see Sect. S1.1), thus having large overlap with the size range most efficiently represented in ACSM. The MMRs from UKESM1 and ECHAM-SALSA are converted into mass concentrations by multiplying the MMRs with the density of the air to facilitate comparisons to chemistry observations given in the units of µg m<sup>-3</sup>. Equivalent black carbon (hereafter BC) was measured with an aethalometer using a cut off diameter of 10 µm (PM<sub>10</sub>). Due to most of the absorbing particles at SMEAR II being at sub-micron range, the difference in the BC mass between PM<sub>1</sub> and PM<sub>10</sub> is only 10 % (Luoma et al., 2019). Therefore, from UKESM1, Aitken and accumulation modes are also used to estimate the total BC. In addition, to obtain SO<sub>4</sub> from H<sub>2</sub>SO<sub>4</sub> (sulfuric acid) which is the UKESM1 native output, a conversion factor is used (see Sect. S1.1). From ECHAM-SALSA, bins with diameters ranging from 19.6 nm to 700 nm (see Sect. S1.2) are used to estimate the total sub-micron OA, SO<sub>4</sub> and BC, including again both soluble and insoluble bins. Here, for ECHAM-SALSA, the largest bin of which a portion also consists of aerosols larger than 1  $\mu$ m (700 nm < d<sub>p</sub> 

Figure 1 ERA-Interim air mass trajectory frequencies for spring (MAM), summer (JJA), autumn (SON) and winter (DJF) are shown in the top row. Frequencies for UKESM1 (e-h) and ECHAM-SALSA (i-l) are shown as differences to the ERA-Interim. Before calculating the differences, the GCM hexagonal grid (150 hexagons in the x-direction) were first regridded to match the gridding in ERA-Interim. Red cross shows the location of SMEAR II.

#### 3.3 Aerosol particle number size distributions

In Figure 2 particle number size distributions from the GCMs are compared with observational data at SMEAR II. The figure reveals that UKESM1 underestimates the number concentration of the small ( $d_p < 50$  nm) particles, especially during summer (Figure 2b, Table S5). This is, however, expected, as the new particle formation from boundary layer nucleation was not implemented in UKESM1 (Mulcahy et al., 2020). ECHAM-SALSA does have a better representation of the PNSD of the smaller aerosol particles during spring and summer when compared to observations (Figure 2c), During warmer seasons, also the absolute number concentrations agree well between observations and ECHAM-SALSA (see nucleation mode from Table S5). This highlights the importance of NPF from nucleation in the boundary layer, especially in summer. During winter, however, ECHAM-SALSA does exhibit some overestimation for Aitken mode aerosols (Figure 2e and Aitken mode from Table S5).

During winter, UKESM1 overestimates larger Aitken and accumulation mode aerosols ( $d_p$  up to 200 nm) compared to the observations (Figure 2b and g), but during spring the number concentration of the accumulation mode aerosols is very close to observations (367 cm<sup>-3</sup> in UKESM1 vs 352 cm<sup>-3</sup> in observations as shown in Table S5). This is somewhat surprising considering the missing growth of small particles from NPF into accumulation mode, however, this could indicate that there are other processes that dominate the accumulation mode. During winter (Figure 2g) the observations exhibit clear bimodal PNSD peaking around 50 and 200 nm but neither of the GCMs is able to capture this behaviour. Overall, both GCMs tend to be shifted towards the larger sizes in all seasons (Figure 2d-g), and this effect is slightly more pronounced in UKESM1. Overall, ECHAM-SALSA better estimates of the peak values of the PNSD, except in winter (Figure 1g), when it overestimates the particle concentrations at the size range of  $d_p = 50 - 100$  nm.

Figure 2 Particle number size distribution at SMEAR II as medians for the day of the year for DMPS measurements (ground level) are shown in (a), followed by the differences between the DMPS observations and the GCMs in (b) and (c). For subplot (c), the measured size distribution was first regridded to the ECHAM-SALSA bins by integrating between the upper and lower limit of each ECHAM-SALSA size bins before calculating the difference. Median PNSDs for each season are shown in (d)-(g) with shaded areas indicating the 25<sup>th</sup> and 75<sup>th</sup> percentiles.

#### 3.3 Chemical composition of the aerosols

392393

Particle chemical composition as a mass concentration for each chemical species from the composition measurements and the GCMs at SMEAR II (trajectory receptor location) is illustrated in Figure 3, and the seasonal patterns are typical for this location. Largest concentration of organic material is present during summer (JJA) and smallest in winter (DJF). Both GCMs also have pronounced OA concentration during summer compared to the other seasons, and UKESM1 captures the pronounced OA concentrations observed during summer particularly well (median OA 2.0 µg m<sup>-3</sup> and 2.2 μg m<sup>-3</sup> in UKESM1 and observations, respectively, Table S6). A portion of the small underestimation of the OA concentrations of the GCMs during spring and summer could, however, be influenced by the height of the observations as chemical composition measurements are conducted at the surface whereas the GCM data shown here are at the trajectory arrival point height at the receptor station (100 m.a.g.l.). Scale difference likely also plays a role, as the point measurements are compared with the GCM grid box values interpolated to air mass trajectories. Monthly data (Figure 3e) shows the second OA peak for the observations to be in February, as expected based on Heikkinen et al. (2020), and in ECHAM-SALSA this peak falls on January. UKESM1 peaks in February, but the difference in the concentrations (compared to observations) between February and January/March is very small. The seasonality of the OA concentrations presented here for both observations and GCMs also agrees with the results from Blichner et al. (2024) who presented the same GCMs but for a different time period. Differences in the monthly peak concentration can be observed for BC too, where observations and UKESM1 peak in February, but ECHAM-SALSA exhibits the largest BC concentrations in January (Figure 3g).

In general, even though a perfect harmonization of the particle chemical composition data between observations and GCMs is not achieved (see Sect. 2.4.3), the median concentrations between observations and GCMs agree relatively well when the overall seasonality is inspected (Figure 3a-d); the concentrations are dominated by OA in all seasons, followed by SO<sub>4</sub> and BC. Inspection of the monthly median concentrations (Figure 3e-g), however, revealed that differences also exist.

Figure 3 Average seasonal mass concentration of sub-micron OA, SO<sub>4</sub> and BC at SMEAR II from the chemical composition measurements, UKESM1 and ECHAM-SALSA is shown in (a)-(d). Black horizontal lines show the median and the boxes extend between 25<sup>th</sup> and 75<sup>th</sup> percentiles. Monthly median (lines) concentrations and 25<sup>th</sup>-75<sup>th</sup> percentiles (shaded areas) are presented in (e)-(g).

# 4 Lagrangian analysis of overall effects of integral precipitation on aerosols at SMEAR II

In this section we use the Lagrangian framework to investigate the potential wet removal of the aerosols. In Sect. 4.1 we first examine the impact of using vertically resolved liquid precipitation (UKESM1 only), which has not previously been done for Lagrangian trajectory analyses. Then we inspect the relationship between accumulated precipitation and aerosols for the two GCM s used in this study: UKESM 1 and ECHAM-SALSA. In Sect. 4.2 we focus on total aerosol mass and number, and in Sect. 4.3 we focus on the OA, BC, and SO<sub>4</sub> portions of the total mass for submicron-size aerosols. Then, in Sect. 4.4, the processes controlling the precipitation-aerosol relationships presented in the previous sections are investigated, and the differences are discussed in detail between the GCMs (Sect. 4.4.1) and within each GCM (Sect. 4.4.2). Supplementary analysis assesses the representability of the models employed here amongst larger group of GCMs (Sect. S4).

# 4.1 Assessment of surface vs. vertically resolved precipitation in Lagrangian wet removal

In earlier studies assessing aerosol-precipitation relationships at SMEAR II using the Lagrangian framework (e.g., Isokääntä et al., 2022; Khadir et al., 2023; Tunved et al., 2013) the vertical position of the trajectories with respect to the precipitating clouds was not considered. The approach, therefore, does not allow for separation between in-cloud and below-cloud precipitation scavenging. Instead, it provides us with the overall effect of precipitation (hereafter noted as wet removal), in which the surface precipitation is used as a proxy for the experienced precipitation by the air mass. This also means that it could include trajectories that travel above the precipitation, potentially confounding interpretation of the results.

For this study, the impact of this simplification was examined by extracting the vertically resolved liquid precipitation from UKESM1, which can be compared to the surface precipitation (see Appendix A). Based on this analysis, it was possible to conclude (see e.g., Figure A1) that for this station the surface precipitation is a relatively good proxy for the experienced precipitation by the air mass. Therefore, and to be able to include the effects due to snowfall, which was unfortunately not extracted with high enough vertical resolution from UKESM1, the surface precipitation is continued to be used in this study. Vertically resolved precipitation was not available from ECHAM-SALSA.

#### 4.2 Relationship between precipitation and aerosol mass and number concentrations

.The removal of the normalized masses (dp = 3-1000 nm, Figure 4a) by accumulated stratiform precipitation for observations and both GCMs exhibit exponential decrease reaching asymptotic behaviour after ~10 mm of accumulated precipitation (after 5 mm for UKESM1 during summer). Normalization of the median mass/number concentration to the median value under zero accumulated stratiform precipitation is used in this study. This approach aims to minimize the influence of differences in the native particle number size distributions (e.g., Figure 1), which affect total mass and number concentrations, and instead highlight the removal attributable to precipitation.

For the particle number concentration (dp = 3-1000 nm), there are clear seasonal differences (Figure 4b). ECHAM-SALSA and the observations show clear seasonal differences in particle number removal, with much more efficient removal in winter than in summer. UKESM1, however, does not display this seasonal contrast—likely because it lacks boundary layer nucleation, a key source of small particles during summer, which leads to similar particle number concentrations across seasons. Inspection of the seasonality is relevant, as differences in the relationships could be driven by different particle size distributions at the station which vary by season due to differences in meteorology (e.g., origin

of air-masses, temperature and sunlight) along the air mass trajectories. Seasonality also impacts to the type of the precipitation (liquid vs snow and stratiform vs convective, for example).

Figure 4c shows that the seasonal patterns (e.g., more samples for smaller precipitation values in summer) in the distribution of accumulated precipitation are similar for both models and observations, thus unlikely to be driving differences in the aerosol-precipitation relationships. The relationships between the aerosol mass, number, and mean stratiform rainfall rate along the trajectory (Figure S6a-b) exhibit similar seasonal differences as the relationships in Figure 4a-b. For example, in summer, UKESM1 exhibits the strongest initial reduction for particle mass (Figure S7a). Observations and ECHAM-SALSA exhibit minimal to no reduction or particle number during summer (Figure S7b), similar to Figure 4b.Non-normalized mass and number concentrations are shown in Figures S7 and S8.

Figure 4 Normalized total ( $d_p$  = 3-1000 nm) particle mass (a) and number (b) at SMEAR II for summer (June, July and August) and wintertime (December, January and February ) as a function of accumulated stratiform surface precipitation (incl. both liquid and snow) along the 96 hour long air mass trajectories for observations (DMPS measurements paired with ERA-Interim trajectories) and GCMs. The coloured points show the median values for each 0.5 mm bin of accumulated precipitation when the number of trajectories in the bin was 10 or larger. The sample size for each corresponding bin is shown in (c).

# 4.3 Relationship between precipitation and aerosol chemical composition

The normalized masses of OA, BC, and SO<sub>4</sub> in submicron-sized particles as a function of accumulated stratiform precipitation (including both liquid and snow) for the observations and the GCMs is shown in Figure 5 (see also Figure S9 showing the same data but grouped differently for easier comparison between the species). The division into warmer and colder months follows the monthly median temperatures (measured at the site) as in Isokääntä et al. (2022). The

454 sample sizes in Figure 5g-h agree well during warmer months between the GCMs. During colder months (Figure 5h) 455 more differences emerge for the smaller precipitation bins (< 3 mm of accumulated precipitation). 456 The general patterns between the observations and GCMs are similar for all species—exponential decrease is observed 457 for the mass concentrations, similar to the relationships between total particle mass and precipitation shown in Figure 4a. 458 The seasonal differences for the total particle mass (Figure 4a) and the chemical constituents are comparable despite the 459 different approach used to separate the data into temperature regimes instead of seasons. During the colder months (Figure 460 5d-f), ECHAM-SALSA exhibits the most efficient reduction for all the three species, as expected based on the reduction 461 of the total aerosol mass (Figure 4a). During the warmer months (Figure 5-c), UKESM1 tends to show more efficient 462 reduction than ECHAM-SALSA, the effect being most pronounced for OA. This is in line with the derived reduction of 463 total particle mass and number during summer shown in Sect. 4.1 (Figure 4a-b), in which ECHAM-SALSA exhibited 464 stronger reduction during winter and UKESM1 during the summer. The observational data presented by Isokääntä et al. (2022) showed that the reduction of SO<sub>4</sub> due to accumulated total 465 466 precipitation in the warmer months was less efficient compared to other species, despite SO<sub>4</sub> being highly hygroscopic and thus relatively easily activated as a cloud droplet. This is relevant also in this study, as the activation into cloud 467 468 droplets followed by precipitation is the dominant reduction mechanisms also for the mass of the different chemical 469 species (discussed in more detail in Sect. 4.4). Similar to Isokääntä et al. (2022), the derived reduction for SO<sub>4</sub> is less 470 efficient (i.e., smaller end concentrations are reached) compared to OA and BC also here for the observations and 471 UKESM1 (Figure S9a-b), though the differences between species are overall smaller but still statistically significant 472 (Kruskal-Wallis rank sum test, p < 0.001). For ECHAM-SALSA, the derived removals between OA and SO<sub>4</sub> do not differ 473 (Figure S9c, Kruskal-Wallis rank sum test, p = 0.2) during warmer months, but BC shows more efficient reduction with 474 the accumulated stratiform precipitation than OA and SO<sub>4</sub>. This could be arising from the fact that, in ECHAM-SALSA, all BC is basically in the soluble particles (Figure S10b) but OA and SO<sub>4</sub> can reside in the insoluble particles as well. 475 476 Isokääntä et al. (2022) hypothesized that the low derived removal efficiency of SO<sub>4</sub> during warmer months could be 477 caused by the species being distributed to different sizes depending on the season. Inspection of the size resolved chemical composition from the GCMs (Figure S10), however, is not able to fully explain the observed seasonal differences: SO<sub>4</sub> 478 479 in the GCMs is almost completely distributed to the soluble accumulation mode, and the seasonal differences are only minor. In ECHAM-SALSA, small contribution of insoluble SO<sub>4</sub> in the accumulation mode is present, but the difference 480

481

482

for the reduction by stratiform precipitation.

16

between the seasons is small (Figure S10b). Other possible explanations could include, for example (but not limited to),

mixing state (internal/external) of the particles and production of SO<sub>4</sub> through cloud processing, which could compensate

Figure 5 Normalized mass concentration for submicron OA, SO<sub>4</sub> and BC at SMEAR II as a function of accumulated stratiform surface precipitation along the 96 hour long air mass trajectories for observations (chemistry measurements paired with ERA-Interim trajectories) and the GCMs for warm (T > 10 °C, (a)-(c)) and cold (T < 10 °C, (d)-(f)) months. The coloured points show the normalized median values for each 0.5 mm bin of accumulated precipitation when the number of trajectories for the bin was 10 or larger. The sample size for each corresponding 0.5 mm bin is shown in (g)-(h).

# 4.4 Process-chain evaluation for understanding the relationship between precipitation and aerosols

To understand the differences between GCMs and observations in Figure 4 and Figure 5, we assess the relative importance of wet removal pathways. Prior studies (Isokääntä et al., 2022; Tunved et al., 2013; Wang et al., 2021), suggests that incloud scavenging, particle activation followed by rainout, is the dominant removal mechanism for submicron particles in this region. For UKESM1 the relative contributions of the removal types (below-cloud impaction, nucleation followed by rainout, and plume scavenging) were quantified using median scavenging coefficients along the trajectories (see Sect. S2). These scavenging coefficients represent the removal within the total atmospheric column, median values along complete trajectories being 0.040 (JJA) and 0.028 (DJF) moles s<sup>-1</sup> for impaction, 0.700 (JJA) and 0.191 (DJF) moles s<sup>-1</sup> for nucleation followed by rainout and 0.001 (JJA) and 0.000 (DJF) moles s<sup>-1</sup> for plume scavenging.

As shown in Figure 6 for organic aerosol (OA), which dominates the particle mass in SMEAR II, e.g., Heikkinen et al., (2020), nucleation followed by rainout dominates removal. Similar patterns are seen for SO<sub>4</sub> (H<sub>2</sub>SO<sub>4</sub>) and BC (Figure S11), supporting that in-cloud removal is the main process in this region, consistent with Isokääntä et al. (2022).

Figure 6 Relative contributions of the different removal pathways in UKESM1 for OA in (a) summer/JJA and (b) winter as a function of time from SMEAR II. Impaction refers to the below-cloud impaction scavenging, nucleation + rainout describes the activation process followed by removal of the particles via the formed raindrops, and plume scavenging is the removal due to convective clouds.

As noted above, nucleation followed by precipitation-driven removal explains the patterns in Figure 4 and 5. To understand differences in this process across models, we compare key variables along air mass trajectories related to incloud removal. Previous studies (Dusek et al., 2006; Ohata et al., 2016; Partridge et al., 2012; Reutter et al., 2009) have emphasized the role of sub-grid processes and variables influencing droplet activation, such as particle size and vertical air motion. We therefore examine how model representations of activation—affected by sub-grid vertical velocities and aerosol size distributions—influence removal.

Key variables controlling the aerosol activation into cloud droplets (presented in Figure 7a-j shows the number of particles with diameter > 80 nm ( $N_{80}$ ) and sub-grid scale vertical velocities (referred as updraught velocities), which control droplet formation. The accumulation mode particles are likely to activate to cloud droplets (Croft et al., 2010; Partridge et al., 2012), and updraught velocities drive supersaturation needed for activation. The activated fraction ( $N_{act}/N_{tot}$ ) is shown in Figure 7k-o, and the rainfall rates (at the surface) are presented in Figure S12. In addition, total number ( $N_{tot}$ ) and total mass of the particles ( $M_{tot}$ ) at the submicron range, a air mass heights and number of activated particles ( $N_{act}$ ) are presented in Figure S13. Chemical composition, relevant for hygroscopicity and droplet formation, is shown in Figure S15.

Together, these factors determine whether the regime is the aerosol- or updraught limited (Reutter et al., 2009). Figure 4 and Figure 5 showed strong seasonal contrasts, and seasonal differences in N80, updraughts, and activation are also evident during transport (Figure 7). Section 4.3.1 discusses seasonal characteristics within each GCM, followed by a model—observation comparison in Sect. 4.3.2.

Figure 7 The evolution of the main drivers for the wet removal (nucleation followed by rainout) along the trajectories. The first row from the top displays the  $N_{80}$  (number of particles for which  $d_p > 80$  nm), the second row shows the sub-grid scale updraught velocities (m s<sup>-1</sup>), third row displays the activated fraction of particles, and the bottom row shows the corresponding trajectory frequencies. For the maps, means are calculated for each hexagonal gridbox (grid resolution being 150 in the x-direction) that the trajectory crosses, and for the rightmost panels, means have been calculated for each hour along the trajectory. For the updraught velocities and activated fractions, only values when trajectory is in-cloud are shown.

# 4.4.1 Seasonal differences within each GCM

In UKESM1, the derived removal for the particle mass during summer is clearly stronger, especially up to ~10 mm of accumulated precipitation, compared to winter (Figure 4a). For the particle number, the differences between summer and winter are less pronounced, and similar concentrations at the receptor station are reached (Figure 4b) with high

accumulated precipitation. A seasonal difference in the absolute values of N<sub>80</sub> can be observed, the number concentration being approximately 100 particles cm<sup>-3</sup> larger during winter compared to summer (Figure 7e). This difference, wintertime values being larger, is also seen in Ntot (Figure S13e). As stated in Sect. 2.2.1, the boundary layer nucleation is absent in UKESM1—a process being especially frequent around SMEAR II during spring and summer (Nieminen et al., 2014). This is likely the cause for the observed differences in  $N_{tot}$  as the model lacks large portion of the smaller particles during summer. For the mass, however, the summertime Mtot is larger (Figure S13j). This could imply that UKESM1 has more numerous medium-sized particles during summer, or, that on average, the particles in summer are larger compared to winter, thus having larger contribution to particle mass. Figure 2 supports the latter scenario, showing the average PNSD at SMEAR II peaking at larger particle sizes in summer compared (~200 nm, Figure 2g) to winter (~100 nm, Figure 2i). The seasonal differences between the updraught velocities in UKESM1 are small, until about 48 hours before arrival (Figure 7j). After that, the summertime updraught velocities exhibit little to no change, but wintertime updraught velocities decrease as the air mass travels closer to SMEAR II. These differences relatively close to the receptor station can be attributed to the geographical distribution of the updraught velocities: close to SMEAR II (across Finland, Sweden and Norway, for example), the values are larger in summertime (Figure 7f) compared to wintertime (Figure 7h). These regions coincide with areas of high trajectory frequency, meaning most air masses pass through them. As a result, the elevated updraught velocities in these regions strongly influence the averages shown in Figure 7j. Activated fractions differ markedly between seasons (Figure 70), with nearly half of aerosols activating in summer compared to about one fifth in winter. These seasonal differences align with the spatial patterns of activated fractions and trajectory frequencies (Figure 7k, p), showing particularly high values over northern Norway and extending into the Arctic Ocean. During winter, the activated fractions in this area are much lower (Figure 7m). The Nact, on the other hand, displays minor differences between the seasons in UKESM1 but is slightly larger in winter. However, considering the fact that Ntot in UKESM1 is much higher in winter (Figure S13e) as mentioned earlier, the larger activated fraction (derived as Nact/Ntot) in summer is reasonable. The chemical composition of particles during their travel in UKESM1 (Figure S14a) reveals that overall, during summer, the mass concentration is completely dominated by soluble modes, whereas in winter, a portion of insoluble OA in the Aitken mode is also present. Soluble SO4 in the accumulation mode contributes more in winter, but this is greatly compensated by soluble OA in both Aitken and accumulation modes during summer. If the higher solubility of OA in summer compensates for the lower SO<sub>4</sub> levels, this could further enhance the particle activation potential in UKESM1 during summer compared to winter. Figure 8 shows the relationship between mean activated fraction and mean updraught velocity that the air mass experienced before arriving at SMEAR II for the summer and winter. For UKESM1, the relationship between these two variables is clearly stronger in summer (slope of 2.12, Figure 8a) compared to winter (slope 0.62, Figure 8b). Therefore, during summer, even a very small increase in updraught could cause a very large increase in the activated fraction. Due to this, the slightly higher updraught velocities during summer, when the air masses approach SMEAR II (Figure 7j), could play a major role, eventually also leading to the larger activated fractions during summer. This, together with the points discussed above, such as the availability of cloud condensation nuclei (CCN), Ntot and particle chemistry along the trajectories, likely causes the seasonal differences observed in the reduction of particle

537538

546547

551552

555556

561562

565566

569570

mass in Figure 4a. When also considering the missing boundary layer nucleation in UKESM1 as mentioned earlier, lack

of seasonality in the derived removal of total particle number in UKESM1 (Figure 4b) can also be explained.

ECHAM-SALSA exhibits stronger reduction (i.e., lower concentrations are reached with increasing accumulated precipitation) during winter than in summer for both particle mass (Figure 4a) and number (Figure 4b). The number of particles for which  $80 \text{ nm} < d_p \le 1000 \text{ nm}$  ( $N_{80}$ ) is relatively similar between summer and winter, exhibiting increase from ~300 up to ~650 particles cm<sup>-3</sup> as the air mass reaches SMEAR II. During summer, the  $N_{tot}$  in ECHAM-SALSA is clearly larger compared to winter (Figure S13e). This is expected due to the strong contribution of small aerosols during summer (e.g., Figure 2c). The total mass ( $M_{tot}$ ), however, is relatively alike between the seasons (FigureS13j), which is reasonable due to the similar contribution of  $N_{80}$  in both seasons, as these particles mostly contribute to particle mass.

The updraught velocities in ECHAM-SALSA exhibit large location-dependent seasonal differences (Figure 7g versus i), especially over the oceans, where the updraught velocities are larger during winter (Figure 7i) than in summer (Figure 7g). However, overall, the average experienced updraught velocities during the transport are rather similar in magnitude between the two seasons (Figure 7j). This overall similarity occurs because the frequency of trajectories passing over the oceans is quite low (Figure 7s) and they therefore do not contribute to the average over all transport directions much. On average, the updraught velocities increase from ~0.4 m s<sup>-1</sup> up to ~0.7 m s<sup>-1</sup> as the air masses approach SMEAR II. Slightly before arrival to SMEAR II (12-36 hours before arrival), difference can be observed in the updraught behaviour: winter updraught starts decreasing around 36 hours before arrival before increasing again at the 12-hour mark. During summer, the updraught increases all the way up ~18 hours, after which it steeply decreases and increases again at the same 12-hour mark as the wintertime updraught. As these differences are taking place relatively close to SMEAR II, it is likely that they are driven by the seasonal differences in the transport and local conditions very close to SMEAR II.

Activated fractions in ECHAM-SALSA display similar trends along their transport, increasing towards SMEAR II, but the seasonal difference in the magnitude is approximately 0.1, wintertime values being larger (Figure 7o). This difference stays nearly constant along the transport. Again, clear seasonal differences within the trajectory transport areas (Figure 7l and n) can be observed, and as the high activated fractions during winter (Figure 7n) do occur in high trajectory frequency areas (Figure 7s), they are more clearly reflected in the values when averaged over all transport directions (Figure 7o). As the seasonal differences  $N_{80}$  in ECHAM-SALSA are negligible, it is unlikely that the number of potential CCN is driving the seasonal differences in activated fractions and in the aerosol mass-precipitation relationships in Figure 4a. When the  $N_{act}$  is inspected (Figure S13t), however, somewhat larger number of particles have activated in winter compared to summer. Thus, when considering the large difference in the total number of particles (Figure S13e), the displayed differences in the activated fractions (= $N_{act}/N_{tot}$ ) are reasonable.

In addition to size, the chemical composition of the potential CCN also has an impact to their activation. The composition of Aitken and accumulation mode aerosols in ECHAM-SALSA (Figure S14b) does reveal, that the particles have relatively similar soluble accumulation mode SO<sub>4</sub> contribution in both seasons. The contribution of soluble OA in the accumulation mode is slightly larger in summer, but during winter, the smaller contribution from OA (in accumulation mode) seems to be compensated by larger contribution from soluble BC in the accumulation mode. Thus, the contribution from soluble modes altogether is relatively similar between the seasons and unlikely causes large differences in the particle hygroscopicity which could impact activation.

In order to investigate whether the seasonal differences in the activated fractions could also be due to slight differences in the sensitivity of activation to updraught velocities, we inspected the relationships between activated fractions and updraught velocities similar to UKESM1. For ECHAM-SALSA, the slope for summer is smaller (slope of 0.18, Figure 8c) compared to winter (slope 0.36, Figure 8b). Thus, during winter, when the updraught increases, the activated fraction

can increase two times as much compared to summer. Therefore, despite the similar number of potential CCN in both seasons ( $N_{80}$ , Figure 7e), larger portion of those activate during winter, resulting to larger  $N_{act}$  (FigureS13t) and activated fractions (Figure 7o). All these findings discussed above are consistent with the stronger reduction for particle mass observed for ECHAM-SALSA in winter (compared to summer) in Figure 4a. During summer, very little to no reduction is observed for the particle number for ECHAM-SALSA in Figure 4b. The particle number concentration, however, is dominated by the small aerosols which are unlikely to activate (see also Figure S13e and Figure 2c). Therefore, even with high accumulated precipitation, no clear reduction is observed in Figure 4b during summer.

Figure 8 Average experienced activated fraction as a function of average experienced updraught velocity along the trajectories. Distribution of the values are shown with the histograms. JJA denotes summer (June-July-August) and DJF winter (December-January-February). Each coloured point denotes a median value determined from a single trajectory. The black lines show the regression line from orthogonal regression applied to the data shown and the legend show the slope, intercept and Pearson correlation (R) between the fit and the data. Note that the black regression lines extend over the whole plot area only due to visualization purposes.

#### 4.4.2 Differences between GCMs and observations

Comparing the two GCMs in Figure 4 it is obvious that the seasonality in the aerosol-precipitation relationships is reversed: UKESM1 exhibits stronger reduction during summer but ECHAM-SALSA in winter. This is unlikely arising from the differences between the intensity of the precipitation during the travel of the air masses, as those are very similar between the GCMs (Figure S12a-e) within each season. However, some of the winter differences may also be attributed to variations in the number of trajectories with specific amounts of accumulated precipitation (Fig. 4c). Observations show a higher frequency of trajectories with low accumulated precipitation (<2 mm), whereas the models produce slightly more trajectories with larger precipitation totals.

**During summer**, UKESM1 has less potential CCN ( $N_{80}$ , see Figure 7e) compared to ECHAM-SALSA, and also the updraught velocities are smaller in UKESM during summer, eventually leading to smaller number of cloud droplets too ( $N_{act}$ , Figure S13t). Comparison of the contribution of different chemical species in the accumulation (as these sizes have

larger contribution to the particle mass) mode (Figure S14, top row), however, reveals that UKESM1 has much larger contribution of the soluble particles. This indicates, that during summer, the particles in UKESM1 have larger hygroscopicity, and could potentially activate more easily compared to ECHAM-SALSA. However, as the resulting Nact (Figure S13t) in UKESM1 is smaller than in ECHAM-SALSA, the potentially larger hygroscopicity in UKESM1 particles do not seem to have significant impact on the droplet formation. When we consider the changes in the PNSD, however, where UKESM1 has significantly less particles but with larger average size compared to ECHAM-SALSA (which has more particles but smaller average size) as shown in Figure 2g and Figure S13e, it is sensible that larger activated fractions are observed for UKESM1 during summer as shown in Figure 7o. The difference in the activated fraction between the GCMs, however, is somewhat larger than what could be expected based on the differences in  $N_{tot}$  and  $N_{act}$  alone. Thus, also the relationships between updraught velocities and activated fractions were inspected to gain further insight. This reveals (Figure 8a and c), that indeed during summer, the slope between activated fractions and updraught velocities in UKESM1 is significantly larger (slope 2.12, Figure 8a) compared to ECHAM-SALSA (slope 0.18, Figure 8c)—difference being over 10-fold. This implies that even a small perturbation in updraught velocity in UKESM1 could increase the activated fraction drastically, resulting in the very high activated fractions observed in Figure 7o, despite UKESM1 having smaller updraught velocities in general. This could indicate a shift in UKESM1 cloud droplet formation from the updraught-limited regime to the transitional regime (e.g., Reutter et al., 2009). These findings align with the stronger reduction of particle mass in UKESM1 as shown in Figure 4a. The reduction of the observed particle mass in summer lies in-between of the two GCMs, initial reduction (up to 5 mm of accumulated precipitation) being more accurately represented by UKESM1.

641

648

651

655

669

656 The differences in the summertime reduction of particle number (Figure 4b) likely arise from the lack of boundary layer nucleation in UKESM1, thus affecting the number concentration of the smallest aerosol particles (see e.g., Figure 2g). As 657 658 already discussed in Sect. 4.4.1, in SMEAR II, NPF is an important source of aerosols and the frequency of the NPF events has significant seasonal variation (Nieminen et al., 2014), summer and spring being most pronounced. Thus, the 659 660 reduction of particle number in UKESM1 during summer (Figure 4b) is similar to the reduction of particle mass (Figure 661 4a), as both are dominated by relatively large aerosols. The summertime reduction of particle number in ECHAM-SALSA 662 coincides with observations, which is to be expected as the Aitken and nucleation mode aerosol concentrations in ECHAM-SALSA are much closer to observed data than UKESM1 (Figure 2g and Table S5). 663

During winter, ECHAM-SALSA exhibits stronger reduction of particle mass compared to UKESM1 after ~5 mm of accumulated precipitation (Figure 4a). The N<sub>80</sub> (Figure 7a-e) is relatively similar between the GCMs, but updraught velocities (Figure 7j) have large difference: UKESM1 updraught velocities range 0.2-0.4 m s<sup>-1</sup>, whereas ECHAM-SALSA has values ranging approximately between 0.5-0.7 m s<sup>-1</sup>. The higher updraught velocities in ECHAM-SALSA likely lead to the larger N<sub>act</sub> (Figure S14t), thus eventually leading to the larger activated fractions for ECHAM-SALSA along most of the transport (Figure 70) due to Ntot being relatively similar between the GCMs (Figure S13e) during winter. It should be noted, that the difference in activated fractions (Figure 7o) far away from SMEAR II is negligible. However, this difference drastically increases when air masses travel to SMEAR II: activated fraction in ECHAM-SALSA continues to increase while UKESM1 fractions stay nearly constant. Thus, it is unlikely that the similar activated fractions far away from SMEAR II significantly impact the reduction observed in Figure 4a.

Comparison of the particle chemistry in the accumulation mode in winter reveals that the GCMs have (Figure S14, bottom row) relatively similar fractions of soluble material. UKESM1 tends to have more SO4, but ECHAM-SALSA more soluble OA and BC. In ECHAM-SALSA, however, the insoluble modes are not strictly insoluble but rather less insoluble compared to soluble modes (Sect. S2.3) and can thus also activate. This could lead to larger  $N_{act}$  (Figure S130) and thus larger activated fraction (Figure 70), considering that the difference in  $N_{tot}$  (Figure S13e) between the GCMs is clearly smaller in winter than what it was in summer. The differences in the relationships between activated fractions and updraught velocities for the GCMs (Figure 8) are more subtle in winter (UKESM1 slope 0.62, ECHAM-SALSA slope 0.36) compared to the values in summertime discussed earlier. Activated fraction in UKESM1 does exhibit higher "sensitivity" for updraught velocities, however, due to the much larger updraught velocities in ECHAM-SALSA, this is likely not enough to increase the activated fraction to the same level, thus leading to less efficient reduction. These assessments align with the particle mass reductions in winter shown in Figure 4a, where particles at ECHAM-SALSA reach slightly lower end concentrations with high accumulated precipitation compared to UKESM1.

The differences in the wintertime reduction of particle number (Figure 4b) are less pronounced compared to those in particle mass (Figure 4a). Initial reduction seems to be more effective on UKESM1, however, after  $\sim$ 5 mm of accumulated precipitation, the reduction in ECHAM-SALSA becomes stronger These differences between the GCMs, however, were not statistically significant (Kruskal-Wallis rank sum test,  $p \geq 0.01$ ). The observational data exhibits stronger reduction than the GCMs during winter for the particle number (Figure 4b) up to  $\sim$ 10 mm of accumulated precipitation. After that, the observations overlap with ECHAM-SALSA. These inconsistencies could also arise from the fact that both GCMs have difficulties representing the bimodal particle number size distribution correctly during the winter months (Figure 2i).

#### 4.4.3 Additional reasons for inter-model differences

Aside from differences driven by aerosol activation, it is important to note that during both summer and winter, additional factors can also contribute to the observed differences in the reductions (Figure 4). For example, the differences in the reduction of the particle mass (Figure 4b) could be influenced by the plume scavenging scheme, a feature only present in UKESM1 (see Sect. S2.4). In this process, aerosol activate into cloud droplets within the convective updraught and fall out via the main precipitation shaft of the cumulonimbus (Kipling et al., 2013; Mulcahy et al., 2020). Note that even though the particle mass is shown as a function of accumulated stratiform precipitation (Figure 4), the air mass trajectories have experienced convective precipitation too. Thus, removal via nucleation (which is more efficient for larger particles) followed by rainout in the convective plume, could also contribute. Inspection of the contribution of the precipitation types reveals that the contribution from the convective precipitation during summer is indeed slightly larger in UKESM1 compared to ECHAM-SALSA (Figure S15). This difference could be reflected in more effective summertime reduction in the particle mass in UKESM1. Another explanation for the more effective reduction of the aerosols during summertime in UKESM1 could be arising from the differences in the parametrizations of the re-evaporation of the falling droplets. In UKESM1, this process is not considered (see Sect. S2.3 and Mulcahy et al., 2020) whereas in ECHAM-SALSA evaporation of the droplets can occur and thus release the aerosols back to the atmosphere (e.g., Stier et al., 2005). During summertime, this re-evaporation could be enhanced due to higher temperatures, leading to less effective observed reduction of aerosols in ECHAM-SALSA compared to UKESM1. However, there can also be other explaining factors, such as location of the precipitation during travel, emissions and dry deposition, which could also indirectly cause differences between the models. Quantifying the exact processes from model parametrizations causing the differences between the observed relationships between aerosol mass and integral precipitation likely requires specific model sensitivity simulations to investigate this, thus being out of the scope of this study.

# 5 Lagrangian analysis on the effects of aqueous phase processing on aerosol chemical composition

In the analysis presented in this section, the relationship between the chemical processing occurring within clouds and fogs in the aqueous-phase is investigated. A special interest is in aqueous-phase SO<sub>4</sub> formation due to its high occurrence in the atmosphere (e.g., Ervens, 2015; Huang et al., 2019; Liu et al., 2020b). We employ a cloud proxy based on relative humidity (RH) along the trajectories similar to Isokääntä et al. (2022). To this end, the history of the air mass is investigated, and if the RH exceeds 94 %, we assume the air mass is in cloud. Further, the air masses were then separated into "clear sky" in which they had no experience of clouds or precipitation during the last 24 hours, and "in-cloud" when the RH exceeded 94 % at least at one trajectory point but no precipitation events occurred during the last 24 hours (Table S7). Only the last 24 hours of the air mass history were considered, as with longer air mass histories (i.e., longer investigated time) the number of strictly in-cloud trajectories decreases due to increasing possibility for precipitation events. Sensitivity tests were conducted by adjusting both the RH limit (from 90 % to 98 %) and trajectory length (from 12h to 60h), but they did not affect our conclusions. It was found that the trajectory length adjustment has large effect on the statistical reliability of the results, hence the investigation is limited to the last 24 hours and thus also stayed consistent with the previous investigation in Isokääntä et al. (2022). This approach is applied for ERA-Interim reanalysis and for the GCM trajectories in similar manner.

- Reader should also note that UKESM1, ECHAM-SALSA and ERA-Interim do not necessarily have identical definitions for RH which could impact the results. To acknowledge this, we also investigated how well the RH along the trajectories actually describes the in-cloud cases by comparing this RH-based proxy to the co-located cloud fraction data from GCMs. This analysis is presented in Sect. S6, and overall, the cloud events (number of the events and their locations at the trajectories) from both approaches were similar, leading to similar conclusions as presented in Sect. 5.1 and 5.2 below. The precipitation used in the classifications here is the total precipitation (including both stratiform and convective precipitation), as aqueous-phase processes are taking place no matter the cloud type. Relative humidity data is from the HYSPLIT output instead of using raw GCM/ERA-Interim outputs with manual co-location. This is because UKESM1 was extracted on pressure levels instead of model levels, and the latter were used in this work for the manual co-location allowing consistency between other variables. The seasonal division applied here is based on the temperature, as in Sect. 4.2. To see whether transport directions and consequently the precursor emissions matter, data is divided into more clean and more polluted air masses (trajectories visiting latitudes below 60° north assigned to polluted sector as in Isokääntä et al., 2022). Trajectory frequency maps for these sectors are shown in Figure S16.
- In this section, the variation in the total submicron mass of different chemical species depending on the experienced conditions is first examined and discussed for the GCMs (Sect. 5.1) and reflected to observations. Then, in the next section (Sect. 5.2), a size-resolved analysis is conducted to determine whether additional insight into in-cloud processing in GCMs could be provided.

#### 5.1 Effects of in-cloud processing for total submicron aerosol mass

Both observations and GCMs show higher SO<sub>4</sub> mass concentrations for cloud-processed air masses within the "cold and polluted" (CP) sector (Figure 9), consistent with findings from Isokääntä et al. (2022). This pattern holds despite the reduced observational dataset due to temporal harmonization with the GCMs (see Sect. 2.4). Other air mass sectors are shown in the supplementary material (Figure S18).

Across all air mass sectors, both GCMs agree well with observations, considering expected differences in the total mass 753 concentrations. Statistically significant increases in SO<sub>4</sub> mass for in-cloud versus clear-sky air masses were found in both 754 observations and models (p ≤ 0.001, Kruskal-Wallis test; Table S8), except for the warm and clean sector (Figure S17g-755 f), where no clear difference was observed. As in Isokääntä et al. (2022), this may reflect limited SO<sub>2</sub> availability for aqueous-phase oxidation in cleaner, warmer air masses. Supporting this, UKESM1 shows the lowest SO2 levels in clean 756 757 sectors (CC and WC; Figure S18e), while higher SO<sub>2</sub> in polluted sectors (CP and WP) coincide with greater SO<sub>4</sub> 758 differences. Recent findings from the Holuhraun eruption (Jordan et al., 2023) also suggest aqueous-phase oxidation 759 dominates SO<sub>2</sub>-to-SO<sub>4</sub> conversion in GCMs. While future increases volcanic activity (Chim et al., 2023), could enhance 760 SO<sub>2</sub> levels and boost in-cloud SO<sub>4</sub> production, ongoing emission controls may reduce anthropogenic SO<sub>2</sub>, potentially 761 counteracting this effect and influencing aerosol size and composition.

The observations shown here do not exhibit statistically significant differences for OA between the clear sky and in-cloud 763 air masses in any of the sectors. The median mass of OA in ECHAM-SALSA is larger for the in-cloud air masses for the 764 cold and polluted sector (Figure 9c and Table S8), but no other sectors exhibit statistically significant differences. 765 However, this difference in the OA mass in the cold and polluted sector is unlikely due to formation of agSOA, as the 766 simulations employed in this study here did not explicitly model the formation of SOA. UKESM1 displays larger 767 differences in the OA mass, in which most are also statistically different. However, the same applies as for ECHAM-768 SALSA, i.e., the model simulations do not include the formation of SOA, and thus the differences must arise from other 769 affecting factors. Both GCMs employ CMIP6 emission datasets as noted in the model setup for AeroCom Phase III GCM 770 Trajectory Experiment, and thus the differences observed here unlikely arise from varying emissions. One should also 771 keep in mind that the representations of OA in the GCMs might differ, and especially their relationship with temperature, 772 relevant driver for SOA formation in general, has been shown to exhibit large structural uncertainties between the GCMs 773 (Blichner et al., 2024).

Isokääntä et al., (2022) did not observe significant aqueous-phase SOA (hereafter, aqSOA) formation from the observations and this has also been noted previously (Graham et al., 2020) for similar boreal environment. Formation of SOA from gaseous precursors dominates this boreal region (see e.g., Petäjä et al., 2022), and thus distinguishing aqSOA from the total formed SOA with our methodology is challenging. For isoprene-dominated environments, the formation of aqSOA is a significant source for total SOA burden (e.g., Lamkaddam et al., 2021). Also biomass burning emissions have been identified as a potential source for aqSOA (Gilardoni et al., 2016; Wang et al., 2024).

It was reported earlier that the observations also suggested increase in the mass fraction of SO<sub>4</sub> when the air masses had been exposed to in-cloud conditions long enough (Isokääntä et al., 2022). To investigate whether similar behaviour could be observed for the GCMs, we calculated the total time spent under the influence of non-precipitation clouds from the 96h long trajectories. Figure 10 demonstrates slight increases in the mass fraction of SO<sub>4</sub> with increasing time spent in non-precipitating clouds for both GCMs. This, however, is somewhat affected by the data size. If inspecting the GCM data which is temporally harmonised to the observations (Figure 10a-b), the conclusion is not as obvious compared to the case were inspecting all available GCM data (Figure 10c-d). This highlights the importance of long enough GCM simulations needed in this type of Lagrangian analysis utilizing single particle air mass trajectories unless ensemble trajectories are utilised.

Figure 9 Median (black horizontal lines and numerical values) particle mass concentrations at SMEAR II with 25th–75th percentiles (boxes) for OA, eBC, and SO<sub>4</sub> for the cold and polluted (CP) air mass sector. The experienced conditions by the air mass are denoted as clear sky and in-cloud (non-precipitating). Subplots include (a) SMEAR II + ERA-Interim, (b) UKESM1 and (c) ECHAM-SALSA.

Figure 10 The mass fractions of OA, SO4, and BC for the more polluted air masses as a function of time spent in in non-precipitating cloud. The top row (a-b) shows the temporally harmonised data and bottom row displays the GCM data without harmonization. The figure shows mass fractions derived from median concentrations for each 1-hour bin.

# 5.2 Effects of in-cloud processing for size-resolved aerosol mass

 To see whether the observed in-cloud formed SO<sub>4</sub> mass in the GCMs (Figure 9b-c) is contributing to same particle sizes as in the observations reported in Isokääntä et al. (2022), the analysis was repeated here for the GCMs. The observations indicated SO<sub>4</sub> mass originating from aqueous-phase processes is mostly contributing to particles with diameters of 200-1000 nm. Figure 11 shows the particle mass concentrations for various size classes derived from the PNSDs from the GCMs for the clear sky and cloud processed air masses for the cold and polluted sector. The three other sectors are shown in Figure S19, and Table S9 shows the results for the GCMs from the statistical significance testing between the clear sky

and in-cloud groups within each size class. Compared to observations, UKESM1 data (Figure 11a and Figure S19) implies the mass increase seems to be mostly distributed to bins with  $d_p = 100-350$  nm and up to 600 nm in the cold and polluted and cold and clean sectors. This is likely due to UKESM1 having large concentrations of particles in general within this size range (see e.g., Figure 2d). Like the observations, UKESM1 does not exhibit any mass increases for any of the size bins in the warm and clean sector (Figure S19e), being in line with no observed increase in the SO<sub>4</sub> mass in the same sector (WC) between the clear sky and cloud processed air masses (Figure S17h).

ECHAM-SALSA (Figure 11b and Figure S19), exhibits increased mass concentrations for sizes starting from  $d_p = 50$  nm (only in cold and polluted sector) up to 1700 nm, depending on the sector. The largest bin here in ECHAM-SALSA might also be influenced by  $d_p = 1$ -1.7  $\mu$ m particles, which are neither considered in UKESM1 nor in the observations when inspecting the chemical components (see Sect. 2.4.2). Like UKESM1, ECHAM-SALSA also does not exhibit mass increases for any of the size bins for the warm and clean sector (Figure S21f).

Figure 11 Median (black horizontal lines and numerical values) particle mass concentrations with 25th–75th percentiles (boxes) for selected size bins for (a) observations with ERA-Interim, (b) UKESM1 and (c) ECHAM-SALSA for the cold and polluted (CP sector). For the latter, the native size bins are shown (bottom row of the legend). The experienced conditions by the air mass are denoted as clear sky and in-cloud (non-precipitating).

An advantage of the GCMs used in this study is their provision of size-resolved chemical composition, shown as mass fractions in Figure S20. For UKESM1, increase in the soluble SO<sub>4</sub> in the accumulation mode can be observed (Figure S20a). Due to the model structure, however, the accumulation mode itself consist of a large spread of particle sizes ( $d_p = 100-1000 \text{ nm}$ ), i.e., internally mixed aerosols with external size modes, thus not providing additional information to our PNSD based analysis. For ECHAM-SALSA, the original sectional bins can be inspected (Figure S20c) thus corresponding to the PNSD bins presented in Figure 11b. All size bins that exhibited mass increases in Figure 11b also exhibit higher mass fraction for SO<sub>4</sub> in Figure S20c.

The observed changes in particle number size distributions (Figure 11) reflect the actual model parameterizations. In UKESM1, SO<sub>4</sub> produced via aqueous-phase chemistry is allocated to the soluble accumulation mode (dp > 100 nm) and coarse mode (dp > 500 nm) (Mann et al., 2010), with the results here showing increases in the 100–600 nm range. In ECHAM-SALSA, aqueous-phase SO<sub>4</sub> is distributed across soluble size bins spanning 50–10000 nm (2a bins; see Table S3, Bergman et al., 2012), with sector-dependent mass increases observed between 50–1700 nm. . In terms of aqueous-phase oxidation of SO<sub>2</sub>, both GCMs have similar parametrizations, and for example, oxidation of SO<sub>2</sub> by ozone (O3) and hydrogen peroxide (H<sub>2</sub>O<sub>2</sub>) is considered in both (Bergman et al., 2012; Hardacre et al., 2021).

#### 6 Conclusions

In this study we investigated the effects of stratiform precipitation (wet removal) and clouds (aqueous-phase oxidation) on submicron aerosols along air mass trajectories. Two global climate models—UKESM1 and ECHAM-SALSA—were analysed using a Lagrangian framework consistent with Isokääntä et al. (2022), now being seamlessly applicable to GCMs (Kim et al., 2020). Our geographical focus was the SMEAR II station in Hyytiälä, Finland, and the surroundings,

representative of the boreal environment.

Our first objective was to investigate whether the trajectory-based relationships between aerosols mass, number and precipitation vary between the observations and the GCMs. For aerosol mass, the derived removal for observations generally fell between those simulated by ECHAM-SALSA and UKESM1 across seasons. This indicates that both models captured the observed mass–precipitation relationship for total aerosol and individual species (OA, SO<sub>4</sub>, BC). In contrast, aerosol number revealed clear model biases that varied by season. In summer, UKESM1 exhibited a pronounced loss of particle number via precipitation compared to both observations and ECHAM-SALSA. This bias likely stems from the absence of boundary layer nucleation, which produces fewer small particles and leaves a larger fraction of particles susceptible to wet removal.

Key variables influencing the wet removal processes, such as number of potential cloud condensation nuclei (N<sub>80</sub>) and updraught velocities, were also examined to evaluate the observed removals. In UKESM1, a strong summer correlation between activated fraction and updraught velocity (Fig. 8) may further increase particle number removal. However, analogous study examining droplet number/CCN versus updraught (Virtanen et al., 2025) show substantial variability across models, highlighting that the relationship. In winter, both models overpredicted particle number removal relative to observations. This overprediction may in part reflect differences in precipitation statistics, with models simulating fewer low-precipitation trajectories (<2 mm) than observed (Fig. 4c). However, other factors such as particle size distributions, activation efficiencies, and limitations in the representation of subgrid-scale meteorology are also likely to contribute. Overall, our results emphasize the need for better representation of particle number size distributions (PNSDs) in GCMs.

Earlier work has indicated that aerosol activation into cloud droplets followed by rainout is the dominant wet removal process. Our results support this, with UKESM1 showing nucleation followed by rainout as the largest contributor. Supplementary analysis comparing a wider ensemble of GCMs indicated that these two models were broadly representative, with their aerosol–precipitation relationships generally falling near the middle of the inter-model spread. Overall, our method using normalized submicron mass and number as a function of accumulated precipitation proved to be effective in comparing removal across models, though it lacks details on particle size evolution—an important topic for future work.

Earlier studies (Isokääntä et al., 2022; Khadir et al., 2023) have noted that surface precipitation data, commonly used in trajectory analyses, may not accurately reflect precipitation experienced by air masses at trajectory height. Here, we used vertically resolved precipitation from UKESM1 and found that surface precipitation serves as a good proxy in this environment, where trajectories largely remain within the mixed layer and stratiform precipitation dominates. However, this analysis only considered liquid precipitation and may not apply to regions where convective precipitation is more prevalent. In such environments, the vertical distribution, intensity, and frequency of precipitation could differ substantially, potentially altering the accumulated wet removal along trajectories. Therefore, while our results are

representative of boreal regions with stratiform precipitation, further work is needed to assess how applicable they are to regions with different precipitation regimes.

Our second objective was to investigate whether the GCMs exhibit similar increase in sulfate mass due to in-cloud production as the observational data. Both GCMs exhibited statistically significant difference in the SO<sub>4</sub> mass when air masses with only clear sky experience were compared to in-cloud processes air masses. The SO<sub>4</sub> mass was larger for the cloud processed air masses for all other air mass sectors (based on temperature and direction) except the warm and clean air masses, where GCMs showed no significant difference between clear sky and in-cloud air masses. These results were consistent with earlier work (Isokääntä et al., 2022). Availability of the SO<sub>2</sub> to be oxidised is likely determining whether we see in-cloud production of SO<sub>4</sub>, and from UKESM1 this was further supported by theSO<sub>2</sub> concentrations and their seasonality. The size-resolved analysis reflected the model parametrizations, the aqueous-phase SO<sub>4</sub> being mostly distributed in the larger aerosol sizes.

As expected based on Isokääntä et al. (2022), we did not observe significant aqueous-phase SOA formation. This is likely due to the studied environment (boreal forest), and has also been noted previously (Graham et al., 2020) for similar boreal forest environment. However, some increases in OA mass were seen in the GCMs despite the fact that aqSOA formation was not explicitly modeled, possibly reflecting other processes or model inconsistencies. A recent study from Blichner et al. (2024) also pointed out the large differences between GCMs concerning their OA-temperature relationships, which could also contribute to the discrepancies observed here.

Overall, both GCMs reproduced the observed exponential decrease in aerosol mass with increasing precipitation and showed similar cloud-processing behaviour for SO<sub>4</sub>. Yet key seasonal differences remain, especially in aerosol–precipitation relationships and their underlying drivers. A primary model bias identified in this study is the difference in aerosol number size distributions compared to observations, particularly the underrepresentation of small particles in UKESM1. Our results suggest that discrepancies arise more from differences in aerosol size distributions and updraught velocities than from the wet removal parametrizations themselves. These variables also affect activated fractions and cloud interactions, and they are shaped by processes beyond the 4-day analysis window.

#### 7 Outlook

While our results show encouraging agreement between observations and GCMs in overall aerosol–precipitation relationships, key differences—especially related to seasonality and aerosol number—highlight the need for further work. Future studies should investigate the evolution of aerosol size distributions along air mass trajectories in more detail and better disentangle gas-phase and aqueous-phase sulfate formation. Expanding analyses to regions with dominant convective precipitation is also important, as the findings here are limited to stratiform, liquid-phase conditions typical of boreal environments. Including a wider range of GCMs, despite the computational demands, would help clarify the structural causes behind the differences observed. Together, these efforts are essential for improving the representation of aerosol–cloud–precipitation interactions in climate models.

#### Appendix A

The lack of vertical resolution in the precipitation data from ERA-Interim reanalysis or Global Data Assimilation System (GDAS, (<a href="http://ready.arl.noaa.gov/archives.php">http://ready.arl.noaa.gov/archives.php</a>, last access: 3.2.2024) in studies using Lagrangian approaches is now being recognised (Dadashazar et al., 2021; Isokääntä et al., 2022; Khadir et al., 2023). Unfortunately, vertically resolved precipitation data from reanalysis datasets or GCMs, with high enough time resolution to be useful for trajectory models, is not a commonly provided diagnostic. For UKESM1, this diagnostic can be extracted. Here, we conducted a comparison between the vertically resolved and surface precipitation data along the air mass trajectories to investigate how well the surface precipitation describes the actual experienced precipitation by the air mass. Only liquid (stratiform) precipitation is inspected, as vertically resolved snowfall was not included in the variable extraction with high enough vertical resolution for this model run.

We first inspected the relationship between the normalized particle mass and number with the accumulated stratiform precipitation, similar to Figure 4. This assessed whether aerosol–precipitation relationships differ between surface and vertically resolved precipitation. Displayed in Figure A1, the results indicate the effects of stratiform precipitation at the height of the air mass are similar to the effects of stratiform precipitation at the surface. This is likely related to the average altitude of the air masses, as for SMEAR II they tend to travel well below the top of boundary layer.

Figure A1 Normalized total ( $d_p$  = 3-1000 nm) particle mass (a) and number (b) at SMEAR II for summer (JJA) and wintertime (DJF) as a function of 0-25 mm of accumulated liquid stratiform precipitation along the 96-hour long air mass trajectories at the height of the air mass (referred as 3D) and at the surface (referred as 2D) for UKESM1. The coloured points show the median values for each 0.5 mm bin of accumulated precipitation when the number of trajectories in the bin was 10 or larger. The sample size for the corresponding bins is shown in (c).

To investigate whether the height of the air mass plays a role, as speculated in Isokääntä et al. (2022), the air mass trajectory altitudes were clustered with Kmeans (e.g., Hartigan and Wong, 1979) and 3 clusters with distinct height

profiles were selected for further analysis. Clustering each season separately provided similar height profiles as clustering of the whole data, and thus the latter approach is presented.

Figure A2 shows the median altitudes of the clusters and the corresponding mean stratiform rainfall rates. Overall, the mean rainfall rates show similar values despite the precipitation diagnostic. In the low-altitude cluster (Figure A2d), overall highest rainfall rates (mean over all trajectories and hours for surface precipitation,  $\sim 0.033$  mm h<sup>-1</sup>) are observed. In the mid-altitude cluster, rainfall rates are smaller ( $\sim 0.016$  mm h<sup>-1</sup>) compared to the low-altitude cluster, and in the high-altitude cluster, the rainfall rates are the smallest ( $\sim 0.010$  mm h<sup>-1</sup>). In the high-altitude cluster (Figure A2f) more differences emerge between the two precipitation types, especially afar from SMEAR II.

Figure A2 Clusters based on air mass trajectory altitudes for UKESM1. In (a)-(c) the black lines show median trajectory altitude as a function of time from SMEAR II and  $25^{th}$  to  $75^{th}$  percentiles are shown with the shaded area. The used arrival height at SMEAR II given to HYSPLIT is indicated with blue horizontal line. The corresponding mean rainfall rates are shown in (d)-(f). Clusters are named based on the maximum altitude the trajectory has resided during the last 4 days. Note the different y-axis limits in subplots (a)-(c).

Each cluster was then further separated by season. The median altitudes, if inspected separately for each season, are nearly identical between the seasons within each cluster, and thus not shown here. Figure S21 shows the differences between the mean liquid rainfall rates between surface and vertically resolves stratiform precipitation (positive difference indicating the rainfall rates at the surface are higher) for each cluster and each season.

During autumn (SON) the two approaches for the precipitation exhibit observable differences only in the high-altitude cluster, where the surface precipitation shows some overestimation of the actual experienced precipitation by the air mass with increasing trend when moving farther away from SMEAR II. This could imply that the air mass has spent some time above or inside the precipitating cloud, as also the air mass altitude increases when moving away from the station (Figure A2a-c). During summer (JJA), all clusters mostly show precipitation at the air mass height being larger than the surface precipitation, expect in the high-altitude cluster (Figure S21c) 72 to 96 hours before arrival to SMEAR II. As the temperatures during summer are higher than in other seasons, this could be indication of evaporation as the surface

precipitation in UKESM1 includes only precipitation that reaches the surface i.e., it is not column integrated. During spring (MAM) and winter (DJF) the surface precipitation shows small overestimation at some points along the trajectories, and the differences are largest at the high-altitude cluster—where, however, the rainfall rates are very small overall (see Figure A2f) for both precipitation types.

#### Data availability

969

975

987

989

- Raw observational data were collected by INAR, University of Helsinki. Field data (particle number size distributions
- and black carbon) are freely available from https://smear.avaa.csc.fi/download (last access: 20 February 2022; Ministry
- of Education and Culture of Finland and CSC, 2022). The ACSM data on aerosol composition are freely available from
- the EBAS database at http://ebas.nilu.no/ (last access: 20 February 2022; NILU, 2022).
- The pre-processed observational data, ERA-Interim and GCMs trajectories along with the co-located variables used in
- this study can be found from Talvinen et al., (2025b).

# Code availability

- Data analysis was conducted in R statistical software (R version 4.2.0, R Core Team, 2019) and Python (version 3.10.4),
- and colour maps for the figures considering colour vision deficiencies were inspired by Crameri et al., (2020).
- The scripts to reproduce the main findings both in R and python can be found from Talvinen et al., (2025a). Python scripts
- for the data conversion (GCM output into ARL) and co-location of the GCM and reanalysis data variables to the
- trajectories can be obtained from DGP.

#### **Author contribution**

- DGP and AV proposed the study. ST, DGP and AV designed the research questions. ST had the lead role in data analysis
- with supporting contribution from PK, DGP, ET and RC. The modelling framework to calculate trajectories from GCM
- meteorological fields was conceived and performed by DGP with support from ZK and JT. The development and
- application of this framework to the AeroCom GCMTraj model submissions was performed by PK with support from
- DGP. Model simulations and data submissions were performed by ET, DGP, TK, EH, HK, TK, DN, DWP, YY, JZ and
- SK. UKESM1 model simulation configuration was supported by AS, and ZK supported the processing of ERA-Interim
- reanalysis data. HYSPLIT trajectories were calculated by PK and the co-location scripts were developed by PK with
- supporting contribution from DGP, ET, ST and RC. Co-location of GCM data and ERA-Interim precipitation to the
- trajectories were performed by ST. Results were interpreted by ST, DP and AV with supporting contribution from all co-
- authors. The manuscript was written by ST with supporting contribution from DP. All co-authors commented, edited and
- gave feedback on the manuscript.

#### Competing interests

At least one of the (co-)authors is a member of the editorial board of Atmospheric Chemistry and Physics.

#### Acknowledgements

- We acknowledge use of the Monsoon2 system, a collaborative facility supplied under the Joint Weather and Climate
- Research Programme, a strategic partnership between the UK Met Office and the Natural Environment Research Council.
- We also thank all the people responsible for the development of UKESM1 and ECHAM-HAM-SALSA. The ECHAM-
- HAMMOZ model is developed by a consortium composed of ETH Zurich, Max Planck Institut fur Meteorologie,

Forschungszentrum Julich, University of Oxford, the Finnish Meteorological Institute and the Leibniz Institute for 995 Tropospheric Research and managed by the Center for Climate Systems Modeling (C2SM) at ETH Zurich. 996 We thank technical and scientific staff from SMEAR II station. 997 DGP would like to extend personal thanks to Ben Johnson and Andy Jones, who provided support for the configuration 998 of the UKESM1 simulations performed as part the AeroCom GCM Trajectory experiment on which these simulations are 999 based. DGP also wishes to thank Hamish Struthers who supported preliminary testing of CAM simulation output during 1000 the development of the coding framework to convert GCM fields into the required format for trajectory calculations, and 1001 Peter Tunved for valuable input and discussions during the development of this framework. 1002 We also wish to thank Eliza Duncan from the valuable input, technical help and discussions during the development of 1003 this work. 1004 Financial support 1005 This work has been supported by European Union's Horizon 2020 research and innovation programme FORCeS (Grant 1006 Agreement No. 821205), Horizon Europe programme via project CERTAINTY (Cloud-aERosol inTeractions & their 1007 impActs IN The earth sYstem, Grant Agreement No. 101137680) and by the project CleanCloud (Grant agreement No. 1008 101137639). 1009 This work has also received support from the Academy of Finland (grant No. 317373 and 317390), Academy of Finland

Flagship funding (grant No. 337550) and the Academy of Finland competitive funding to strengthen university research

profiles (PROFI) for the University of Eastern Finland (grant No. 325022 and 352968).

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
