# Peer review of "Towards an improved understanding of the impact of clouds and precipitation on the representation of aerosols over the Boreal"

_EGUsphere, 2025_

## Referee Comment (RC2)

**Review of manuscript egusphere-2025-721**

**Summary**

The authors examine how clouds and precipitation influence aerosol lifecycles in a boreal forest in Finland. They conduct simulations with the two general circulation models UKESM1 and ECHAM-SALSA. The aerosols are represented with a modal scheme in UKESM1 and a sectional scheme in ECHAM-SALSA. They analyze their simulations in an Eulerian and Lagrangian manner and evaluate their results against in-situ observations. They find seasonal differences in the aerosol-precipitation trends of UKESM1 and ECHAM-SALSA. The paper provides valuable insights but should be revised before publication. There are two general comments and some specific commments.

**General comments**

- The authors should present their study in a more concise way. In its current form, the main part spans 49 pages and the supplemental part spans 26 pages. A more concise presentation would make it easier to grasp the main ideas.

- The authors often present normalized or relative values, for example, in figures 3, 4, 5, and 6. The author should provide absolute values instead or in addition. That would make it easier to interpret their results.

**Specific comments**

- Please make the font sizes in figures 1, 3, 4, 5, 9, and 11 a bit larger.

- In figure 6, please provide absolute numbers for removal pathways.

- In figure 7, please show the trajectory frequencies in separate, larger panels to better show of the spatial details.

- In figure 8, please show a histogram instead of a scatter plot to better show the different frequencies.

- Please show the time series in figures 3 and A2 with line instead of bar plots.

---

## Referee Comment (RC3)

**Towards an improved understanding of the impact of clouds and precipitation on the representation of aerosols over the Boreal Forest in GCMs**

**Manuscript overview**

This manuscript presents research that investigates aerosol-cloud interactions at a Boreal forest location. The study focusses on how cloud and precipitation processes affect aerosol properties including aerosol concentration, size distribution, and chemical composition. The study compares model output from two global climate models to observations made at the Boreal forest location. The study then uses a Lagrangian framework to investigate the aerosol cloud interactions along trajectories prior to their arrival at the measurement site in summer and winter conditions.

The study presents detailed results describing aerosol-cloud interactions in the Boreal forest location and provides useful insights into the global climate model's behaviour, for which the authors are to be commended. However, the manuscript requires improvements so that the results, and their significance, are clear to the reader. I have made some general and technical comments below.

**General Comments**

The manuscript presents quite a lot of detail across the results sections, supplementary material and the appendices, and I found it difficult to follow and understand. I think that the results of the research need to be better focussed on the study's main objectives (1-2).

For me, the novel aspect of this study was the model-model-observation comparison, why there are differences between the models, where there are model biases compared with the observations and how the models might be improved. I think this 'story' needs to be made clearer throughout the manuscript.

1. I think the additional research questions (a, b) detract from the focus of the manuscript. The relevant results could be reported in the supplementary material. I also do not think additional research question (b) is necessary. This is because the analyses only looks at if precipitation in UKESM1 and ECHAM-SALSA is representative of AeroCom GCMTraj. However, in this study aerosol behaviour in the models is also very important for explaining differences between the models and observations.
2. I think that the manuscript could be better organised to help the reader understand the main results and their significance.
   a. I suggest that the authors decide on what order to present their results to best tell their story and keep this consistent throughout the manuscript. This should then be summarised at the end of the Introduction to guide the reader through the manuscript. E.g. "In Section 2 we present…." etc , etc.
   b. Sections and sub-sections could also be organised more clearly to guide the reader. For example, in the Introduction, the Lagrangian analysis is discussed across Paragraphs 1 and 3.
   c. The manuscript's supplementary material is very useful to understand some of the study details. However, I think that the main manuscript, including figures, needs to 'standalone'. For example, where results have been presented for one aspect of the analysis (e.g. for one season) a summary of the rest of the analysis and reference to the Supplementary materials could be given at the end of the section.
   d. Some longer sub-sections could be split up further. For example by aerosol type (organic aerosol and sulphate) in Section 4.2.

2. I think the models should be referred to as global climate models or Earth System models (ESMs). UKESM is definitely an Earth system model.
3. Isokääntä et al., 2022 is regularly referenced. This manuscript appears to use observational data from Isokääntä et al. and extend their analysis techniques to GCMs. In the Introduction the authors could consider including a paragraph clearly summarising of Isokääntä et al. and how this study builds on it.

**Technical comments**

General technical comments

- Colloacation => co-location
- Airmasses => air masses
- Use of 'hexagon'. Is this analogous to a model grid box?
- Please consider using "particles cm$^{-3}$" instead of "# cm$^{-3}$".
- For clarity please consider using "updraft velocity" rather than "updraft".

Abstract

- This needs to include the main study objectives and the main conclusions, i.e. what does the study tell us about how precipitation impacts aerosol-cloud interactions in GCMs?

Introduction

- L53: **and therefore** partly masking….
- L67: "Such frameworks facilitate  the development…" OR "Such frameworks **pave** the way for the development…."
- L85-117: Paragraph 3 is very long and would benefit from being broken up.
- L126-129: *"In this work, the effects of wet processing (wet removal and aqueous phase processing) along air mass trajectories on modelled aerosol size distributions are compared with long-term observations of aerosol size distributions in Hyytiälä, Finland. The observations are combined with ERA-Interim reanalysis trajectories, and the trajectories to be utilized with the GCM variables are calculated with the GCM simulation (nudged to ERA-Interim reanalysis) output meteorology"*

  These sentences are really important because they describe the research, but the second sentence is difficult to understand.

- L98-99: *"…more efficient removal on the number concentration of the accumulation mode particles with increasing accumulated precipitation…"*

  Suggest: "…more efficient removal of the accumulation mode particles with increasing accumulated precipitation…"

- L100: environment => environments

Methods:

- L168-169: "*The dataset used in this study is reduced compared to Isokääntä et al. (2022) and extends to the end of 2018 to facilitate comparisons to the simulation period of the GCMs.*"

  Do you mean a shorter time series? Or have other constraints been applied to the data for quality control purposes? What is the time period used for this study – is it 2005-2018?

- L174-178: This sentence needs rewriting for clarity. I suggest describing one model component per one sentence.
- L181-185: Please edit this sentence for clarity. I suggest using shorter sentences.
- L207-208: "…chemical components including mass mixing ratios of sulfate noted here as SO4 (extracted as sulfuric acid H2SO4 and then converted, see Sect. S1.1),…" See also Section 2.4.3.

  Could the authors please explain why they have taken this approach? UKESM does output mass mixing ratios of sulphate aerosol ($SO_4^{2-}$) for the different aerosol modes (nucleation, Aitken etc) and I'm not sure why these were not used. UKESM's $H_2SO_4$ diagnostic is sulphuric acid in the gas phase.

- Section 2.2: Could the authors please include a mention of how were the GCM simulations extended beyond 2014 and up to 2018? Was a future simulation used from one of the SSP scenarios used? Although there is little expected difference in SSPs for the 2014-2018 period, it would be useful to see what was used.
- L241 By "4 d" do you mean four day?
- L277: "…allows to retain the…" => "…permits retention of the…"
- L277-279: While I understand the reasons for the author's approach here, I'm not sure 'ensuring consistency' is the right term because the precip is ERA, which the aerosol variables are GCM.
- Section 2.4.1: Were the hourly observations averaged to co-locate with the 3 hourly model data?
- L300: Please define DMPS here
- L300: Please define ACSM here

Results sections

- In general, at the start of a sub-section or paragraph, please try to give an overview of what the relevant figure is showing. The sub-figures can then be referenced as each result is described. Please see the Section 3.1 comments for a suggestion.

Section3:

- In paragraph 1 consider indicating which variables are be compared between the model and observations - Aerosol particle number size distributions and chemical composition
- L340-342: "For the GCMs, fewer studies looking into aerosol properties at single sites exist, but Leinonen et al. (2022), for example, conducted an extensive study comparing long-term aerosol particle seasonality and trends in observations and GCMs in multiple locations, also including SMEAR II."

  This could be in the Introduction so that the Results sections can focus on the study's analyses

- L347-354: I suggest that this could be a short subsection. Something like "Comparison of transport for ERA and GCMs". Figure S4 could be included in the main manuscript with the GCM output shown as difference plots compared with the ERA data. This would also help illustrate the study location.

Section3.1

- L356-363: I suggest re-writing the first paragraph as follows to help guide the reader. Sub-figures can be referenced as the results are described. Information about the time periods plotted and which percentiles are used for the statistics can be clearly stated in the Figure caption.

  In Figure 1 particle number size distributions from the GCMs is compared with observational data at the SMEARII. Median aerosol number concentrations for nucleation, Aitken and accumulation mode are shown in Table S5. ECHAM360 SALSA data in Figure 1c is presented in its native resolution for size bins falling between $dp = 3.0 - 1700$ nm and those size bins are positioned within the y-axis to the geometric mean of the ECHAM-SALSA size bins (see Table S3). To calculate the difference in Figure 1e, the measured size distribution is regridded to the ECHAM-SALSA bins by integrating between the upper and lower limit of each ECHAM-SALSA size bin.

- L375: *"…this could indicate from other processes which dominate the accumulation mode."*

  I suggest re-writing as: "…this could indicate that there are other processes that dominate the accumulation mode."

- Figure 1: Consider reducing the number of sub-figures. For example, omit 1b and 1c and just show the difference plots (1d and 1e).

Section 4

- I suggest that Section 4 is re-written to help guide the reader through it and to help the reader understand the main results. I suggest that the analysis of the impact of extracting the vertically resolved liquid precipitation from UKESM1 described in paragraphs 1 and 2 is described in its own section, e.g. 4.1. I then think that including the analysis of the representativeness of precipitation in UKESM and ECHAM-SALSA compared to the AeroCom GCMTraj ensemble here is distracting. I think this analysis, together with Appendix B, could be moved to the supplementary material.
- L440: "smallest" => smaller
- L445-451: I suggest removing the sentence "*The analysis is simplified by removing the size dependent component noted in previous literature (see e.g., Figure 3 in Isokääntä et al., 2022 and Figure 4 in Khadir et al., 2023)…*"

  I don't understand what the sentence is referring to here, but it could be included when the relevant results are shown later.

- L445-451: Based on the above comments I then suggest starting Section 4 with the following paragraph:

  "In this section we use a Lagrangian trajectory framework to…. In Section 4.1 we examine the impact of using vertically resolved liquid precipitation, which has not previously been done for Lagrangian trajectory analyses. Then the investigation inspects the relationship between accumulated precipitation and aerosols for the two GCMs used in this study: UKESM1 and ECHAM-SALSA. In Section 4.2 we focus on total aerosol mass and number. In Section 4.3 we focus on the OA, BC, and SO4 portions of the total mass for submicron-size aerosols. Then, in Section 4.4, the processes controlling the precipitation-aerosol relationships are investigated, and the differences are discussed in detail between the GCMs and within each GCM."

- Section4.1: To help the reader understand the results in this section I think it would be helpful to describe them after Figure 4. Why you have looked at seasonal periods (as in L466-474) can then be discussed as you discuss the results shown in Figure4. Additional results such as those in Figures S7-9 can be mentioned at the end of the Section. I also think that statements like *"Non-normalized mass and number concentrations are shown in Figures S8 and S9, and those are similar to Isokääntä et al., (2022) which employed whole year data and total precipitation"* (L475-476) are not useful. How are the mass and number concentrations similar to Isokääntä et al., (2022) and why is this important? The novelty of this study is the model-observation comparison. If it is important the observational data used here is similar to Isokääntä et al., (2022), there should be an additional analysis.
- L471-474:
  (i)     "…which further cause differences…" => "…which causes further differences…"
  (ii)    I suggest splitting this sentence up for readability.
- L477: In describing Figure 4 here and elsewhere in the manuscript the authors refer to "removal of the normalized masses…" However, the plots are not actually showing removal processes. I think it would be more correct to refer to e.g. reductions in normalized masses.
- L480-483: Why would there be a seasonal effect of UKESM's missing boundary layer nucleation process?
- L539 and Figure 5 caption: "…number of data rows for the bin was 10 or larger." What does this mean?
- L575: Activated fraction? Is this the fraction of $N_{80}$ particles that have been transported upward?
- L609-610 "These regions coincide with the high trajectory frequencies; thus, the high updraughts are being reflected on the averages irrespective of transport direction in Figure 7j."

  Could the authors please clarify this statement.

- L614: "…where the north of Norway…" The region of very high activated fraction appears to extend over the Arctic Ocean.
- L621-623: *"Assuming the solubility of OA compensates for the missing portion of SO4 in summer, these differences could increase the particle activation potential even further during summer in UKESM1 (compared to winter)."*

  I think this is a bit difficult to follow.

- L624-626: Please consider re-wording as "Figure 8 shows the relationship between mean activated fraction and mean updraft velocity that the air mass experienced before arriving at SMEAR II for the summer and winter periods."
- Figure 8: Please consider resizing the text annotations on the plots.
- L634: Here, I think it should be "lower concentrations" rather than "smaller concentrations"
- L649: "…which is steeply…" => "…which it steeply…"
- L650-651: *"As these differences are taking place relatively close to SMEAR II, it is likely that they are driven by the seasonal differences in the transport very close to SMEAR II."*

  Could the authors please explain why that might be?

- L722: I suggest re-wording *"…UKESM1 updraughts are below and above 0.2 m s-1…"* to "…UKESM updraught velocities range from 0.2-0.4 m s$^{-1}$…"
- L749-768: Please consider putting this text in a new sub-section with a heading something like "Other reasons for model differences"

- L826-829: Possibly, but this would be balanced against likely decreases in SO2 emissions due emission control policies in many regions.
- L837-846: Please consider putting this paragraph before the preceding discussion on SOA
- L835-834: In this study region or in general?
- L844-846: Are the same (e.g. CMIP6) emissions data sets being used in both models? Would rule out differences in emissions.
- Figure 11: I think that it would be useful to include the observations for the for corresponding period and sector.
- L896-904: Please consider re-wording this paragraph because I cannot understand what it means.
- L902-904: Both models have very limited aqueous phases schemes – do the results imply that this does not matter too much, at least at the Boreal forest location?

Section 5

- In common with Section 4, there are many references to the supplementary materials. I think the text could generally be improved by clearly summarizing the main findings as shown in the main manuscript's figures.

Conclusions and outlook

- I suggest that this section is edited to ensure that the main results and their impact is clear to the reader. I also think that the authors should include some discussion on the impact of their results for GCMs. For example:
    - What were the main model biases against observations?
    - Are there any processes that should targeted for model development?
    - Are the results likely to be applicable to different locations? This is difficult, but important because GCMs are used for global climate model prediction.

Supplementary

- Tables S1 and S3: I'm missing a description of "GSD" and "GMD" in column 4.
- Figures S4 and S5: Difference plots would help to illustrate the differences, or lack thereof, between ERA and the model simulations.

---

## Author Comment (AC1)

**Referee comments in black, author replies in blue**

**Referee #1**

The authors compare output from two general circulation models that were nudged to reanalysis conditions against aerosol measurements taken at a boreal forest ground site. The analysis includes both Eulerian and Lagrangian perspectives. The latter is used to infer if aerosol sinks and sources are plausible in both models. The manuscript is of great relevance and should be published after major revisions.

**Major concerns**

I think the manuscript (not considering appendix and supporting material) is too long and should be more concise in style. A good example is the "Conclusion and Outlook" section, which spans over 2 pages. While there are key conclusions in this section, it is hard to find them when mingled with discussion material. Instead, the authors could split this section into a conclusion sections (as also suggested by the ACP guidelines) that is very concise and an outlook (or else discussion) section. In the same manner, the authors should attempt to shorten sections 3-5, by perhaps moving discussion-style material to a dedicated discussion section.

We appreciate the reviewer's comment regarding the length of the manuscript. We also noted that other reviewer raised similar concerns. In response, we have thoroughly revised the manuscript and significantly shortened it by streamlining the presentation and condensing sections where appropriate. The result sections 3-5 have been shortened, and some text has been made into its own subsections as suggested by another referee. We hope this has further clarified the results and also made some of the sections less text-heavy. We have decided to keep the discussion material intermingled with the results (essentially "results and discussion") as we consider it provides more clarity to discuss the results when they are first presented. However, we have now reduced the text in the conclusion section and the conclusion and outlook sections have been separated into different sections.

We recognize that the manuscript still remains relatively long; however, after careful consideration, we believe that further reductions would risk omitting key information and undermining the clarity or completeness of the work. Prior to submission, we also considered the option of splitting the manuscript into two separate papers. However, this approach felt somewhat artificial given the tight integration of the analysis across datasets, models, and methods, and we felt that dividing the content would have disrupted the logical flow and weakened the overall message of the study.

We hope the revisions strike a better balance between completeness and readability, and we thank the reviewer again for raising this point.

Throughout the Lagrangian analysis the authors mostly show median values. I'm wondering how large the interquartile range is and whether the main conclusions are affected when considering this range.

We thank the referee for this insightful comment. The use of median/mean values throughout the Lagrangian analysis is intentional for several reasons. Many of our figures present multiple models alongside observations (e.g., Figures 4 and 5 and SI Figure S4) or multiple seasons (e.g., Figure 7), where including interquartile ranges (IQRs) (or other metrics describing the variability) would significantly clutter the visuals and obscure the primary trends we aim to highlight. Averages effectively summarize the central tendency relevant to our main conclusions.

To address the referee's concern, we have prepared versions of Figure 4 that include IQRs, and part of Figure 7 with standard deviations included. These versions illustrate that while adding IQRs (or

standard deviations) provides additional variability information, it does not alter the key patterns or conclusions. We find that presenting all models together with IQRs/standard deviations (see Figure 1 and Figure 3) makes the figures difficult to interpret due to overleaping , and separating each model into subplots (see Figure 2 below), although clearer, results in an excessive number of figures that would reduce the manuscript's conciseness.

In summary, although the IQRs and standard deviations provide additional information and context, their inclusion does not affect our main findings, and we believe that the current presentation best balances clarity and completeness.

[Figure]

*Figure 1 Example of manuscript Figure 4a displaying the normalised mass concentrations as a function of accumulated precipitation. The points denote the normalized medians and the lines indicate the normalized IQR ranges.*

[Figure]

*Figure 2 Example of manuscript Figure 4a displaying the normalised mass concentrations as a function of accumulated precipitation similar to Figure 1 above, but now each model/season is separated into its own subplot for clarity.*

[Figure]

*Figure 3 Example of manuscript Figure 3 – time evolution of selected variables. Lines show the mean values while the shaded areas display the standard deviations.*

**Minor concerns**

l. 57 Please define "sectional approaches".

*We have now reformulated the referred sentence to such that it includes the requested definition: "Traditionally, discrepancies in particle size distributions between observations and models exceed those between modal and sectional approaches, with sectional methods dividing the distribution into discrete size bins (Mann et al., 2012). However, larger differences in concentrations may emerge when chemistry of the aerosols is inspected (Laakso et al., 2022)."*

ll. 428ff (and Fig. 3). Why is rainfall rate increasing closer to the site?

*While we cannot provide a definitive explanation for this pattern, we hypothesize that it is most likely related to local meteorological conditions. We observe that the Hyytiälä region typically receives more rainfall compared to the surrounding areas, and the effect is consistently present across all models (see Supplementary Fig. S4) and in ERA-Interim, so no single model stands out in this regard. The spatial distribution of precipitation (see Supplementary Fig. S12) also aligns with the referred figures: closer to the site, the average rainfall is higher. Very high rainfall rates along the coast of Norway could additionally contribute, as trajectories arrive from there relatively often.*

We note that this analysis is based on surface precipitation, and since the trajectories travel at lower altitudes near the site, they may experience more rainfall compared to those farther away and higher up (when assuming high-altitude trajectories could in some cases, be above the cloud base). This effect is also somewhat visible in the trajectory analysis presented in the Appendix, where the low-altitude cluster shows slightly higher mean rainfall rates compared to the high-altitude cluster (Figure A2). However, this is unlikely to be the dominant factor, as rainfall at the trajectory heights is similar to surface precipitation (i.e., trajectories are most likely below the cloud base much of the time). Taken together, the most plausible explanation is location-dependent variability and local meteorological conditions.

ll. 805ff. Is there a chance that "in-cloud" and "clear-sky" groups correspond to very different regions (with unique local aerosol sources)?

The referee raises an important point, which we carefully examined. To ensure comparability, we inspected trajectory sources across the classifications and sectors used in our analysis (see Supplementary Fig. S16). This step confirmed that observations and models are compared on an equal basis, with trajectories arriving from the same directions. In Figure below (Figure 4) we further show 24-hour back trajectories for ERA-Interim as an example, separated into clear-sky and in-cloud groups (definitions in Table S7). The trajectories arrive from similar directions within each sector, also regardless of classification, making it unlikely that transport direction explains the sulfate signal. While the in-cloud group contains fewer trajectories by definition, our sensitivity tests (Sect. 5), which also alter number of trajectories in each group, confirmed that group size does not affect the results.

Taken together, the analyses demonstrate that our trajectory grouping is consistent and robust. Similar transport directions across groups, combined with sensitivity tests on trajectory number, confirm that neither factor drives the sulfate signal. We are therefore confident that the observed sulfate differences reflect genuine atmospheric processing.

[Figure]

[Figure]

*Figure 4 Air mass trajectories for ERA-Interim sectors (CP, CC, WP and WC), shown separately for clear sky and in-cloud trajectories.*

**Typos/style**

ll. 923-924 Please check this sentence.

The referred sentence is now rewritten.

l. 943 "extremeties"

The referred wording has now been fixed. Please note that the text has been removed from the main manuscript as the corresponding analysis was moved to supplementary material (see supplementary section S4) to keep the focus on the UKESM1 and ECHAM-SALSA models.

**Citation**: https://doi.org/10.5194/egusphere-2025-721-RC1

**Referee #2**

Summary

The authors examine how clouds and precipitation influence aerosol lifecycles in a boreal forest in Finland. They conduct simulations with the two general circulation models UKESM1 and ECHAM-SALSA. The aerosols are represented with a modal scheme in UKESM1 and a sectional scheme in ECHAM-SALSA. They analyze their simulations in an Eulerian and Lagrangian manner and evaluate their results against in-situ observations. They find seasonal differences in the aerosol-precipitation trends of UKESM1 and ECHAM-SALSA. The paper provides valuable insights but should be revised before publication. There are two general comments and some specific comments.

General comments

- The authors should present their study in a more concise way. In its current form, the main part spans 49 pages and the supplemental part spans 26 pages. A more concise presentation would make it easier to grasp the main ideas.

We appreciate this important comment. While the manuscript has not become substantially shorter (word count has been reduced from ~15500 to about 13900), we have focused on improving readability through clearer wording, better organization, and removal of redundancies. Our aim was to retain all technical details necessary for transparency and robustness, which naturally take space, but to present them in a way that is easier to follow. In response to this and similar feedback from other reviewers, we have reorganized several sections to highlight the core findings more effectively, and we have moved Appendix B to the supplementary material.

- The authors often present normalized or relative values, for example, in figures 3, 4, 5, and 6. The author should provide absolute values instead or in addition. That would make it easier to interpret their results.

We chose to use normalized values because the absolute concentrations differ significantly between models and between models and observations, primarily due to large variations in number concentrations. Using absolute values in the main figures (e.g., Fig. 4) would obscure the differences we aim to highlight. Our objective was to evaluate wet removal efficiency consistently across models, focusing on relative—not absolute—removals. In other words, we were interested in the steepness of the removal curve, normalized to each model's or observation's baseline concentration, rather than in comparing total removed mass or number.

To address potential interest in absolute values, we had included these in the Supplementary Material for some of the main figures (for Fig. 4 -> Figures S8 and S9). These figures also illustrate the challenge posed by the wide range of absolute concentrations across models. We would not be able to compare the "derived removal" across models if we would use absolute values, as the differences in the absolute concentrations would obscure the differences we are interested in. Regarding Figure 6 which displays the contributions of the removal types, we were only interested in the contributions of these different removal processes— i.e., which process contributes most to the mass removal of the particles. Regarding this plot, however, we have now added the average values to the text as the referee requested in the specific comments to give an overview on how large absolute removals are in question.

Specific comments

- Please make the font sizes in figures 1, 3, 4, 5, 9, and 11 a bit larger.

We thank the referee for noting this, as the font indeed appears relatively small in some of the figures. We have now improved the fonts and figure sizing for better readability.

- In figure 6, please provide absolute numbers for removal pathways.

We have now added the average removal values, during travel, to the text for both summer and winter and for all removal types: "*These scavenging coefficients represent the removal within the total atmospheric column, median values along complete trajectories being 0.040 (JJA) and 0.028 (DJF) moles s$^{-1}$ for impaction, 0.700 (JJA) and 0.191 (DJF) moles s$^{-1}$ for nucleation followed by rainout and 0.001 (JJA) and 0.000 (DJF) moles s$^{-1}$ for plume scavenging.*"

- In figure 7, please show the trajectory frequencies in separate, larger panels to better show of the spatial details.

The panel size for the trajectory frequencies (Figure 7p-s) have now been increased.

- In figure 8, please show a histogram instead of a scatter plot to better show the different frequencies.

We thank the referee for this suggestion, as it gives a nice overview on how the values are distributed. The histograms of activated fraction and updraughts have now been added to Figure 8.

- Please show the time series in figures 3 and A2 with line instead of bar plots

The bar plots have now been changed to line plots in the referred figures. Please note that Figure 3 has been moved to supplementary material to accommodate the feedback from the other referees.

**Referee #3**

**Manuscript overview**

This manuscript presents research that investigates aerosol-cloud interactions at a Boreal forest location. The study focusses on how cloud and precipitation processes affect aerosol properties including aerosol concentration, size distribution, and chemical composition. The study compares model output from two global climate models to observations made at the Boreal forest location. The study then uses a Lagrangian framework to investigate the aerosol cloud interactions along trajectories prior to their arrival at the measurement site in summer and winter conditions.

The study presents detailed results describing aerosol-cloud interactions in the Boreal forest location and provides useful insights into the global climate model's behaviour, for which the authors are to be commended. However, the manuscript requires improvements so that the results, and their significance, are clear to the reader. I have made some general and technical comments below.

We are very grateful for the referee regarding the feedback for our work. We appreciate the fact that the referee clearly has taken the time to go through the manuscript, despite its length. We hope that all the adjustments we have now made have significantly improved the manuscript and it will be easier to grasp for future readers.

**General Comments**

The manuscript presents quite a lot of detail across the results sections, supplementary material and the appendices, and I found it difficult to follow and understand. I think that the results of the research need to be better focussed on the study's main objectives (1-2).

We thank the reviewer for this helpful comment. We acknowledge that the original manuscript included a substantial amount of detail, which may have made it challenging to follow the main storyline. In the revised version, we have carefully restructured the results sections to place greater emphasis on the study's main objectives (1–2). We also removed or condensed content that was not essential to the core message and ensured that the supplementary material now supports rather than distracts from the main findings. These revisions are aimed at improving clarity, coherence, and overall readability, and also to condense our work for accessibility.

Please see our point-by-point responses regarding the more detailed feedback below.

For me, the novel aspect of this study was the model-model-observation comparison, why there are differences between the models, where there are model biases compared with the observations and how the models might be improved. I think this 'story' needs to be made clearer throughout the manuscript.

1. I think the additional research questions (a, b) detract from the focus of the manuscript. The relevant results could be reported in the supplementary material. I also do not think additional research question (b) is necessary. This is because the analyses only looks at if precipitation in UKESM1 and ECHAM-SALSA is representative of AeroCom GCMTraj. However, in this study aerosol behaviour in the models is also very important for explaining differences between the models and observations.

We thank the referee from this suggestion, and we have now removed the "additional research questions" from the introduction. Instead, we now refer to these "robustness checks" in the relevant parts of the results. This also helped to condense the work overall without sacrificing key information.

2. I think that the manuscript could be better organised to help the reader understand the main results and their significance.

    a. I suggest that the authors decide on what order to present their results to best tell their story and keep this consistent throughout the manuscript. This should then be summarised at the end of the Introduction to guide the reader through the manuscript. E.g. "In Section 2 we present…." etc , etc.

    b. Sections and sub-sections could also be organised more clearly to guide the reader. For example, in the Introduction, the Lagrangian analysis is discussed across Paragraphs 1 and 3.

    c. The manuscript's supplementary material is very useful to understand some of the study details. However, I think that the main manuscript, including figures, needs to 'standalone'. For example, where results have been presented for one aspect of the analysis (e.g. for one season) a summary of the rest of the analysis and reference to the Supplementary materials could be given at the end of the section.

    d. Some longer sub-sections could be split up further. For example by aerosol type (organic aerosol and sulphate) in Section 4.2.

We appreciate the reviewer's detailed feedback. To specifically address these suggestions, we have revised the Introduction to include a brief summary of the manuscript structure, guiding the reader through the content (e.g., "In Section 2 we present…"). We have also improved the internal consistency in the ordering and flow of the results, reorganized the discussion of the Lagrangian analysis in the Introduction for clarity, and split longer sub-sections (e.g., Section 4.2) where appropriate. Regarding the seasons (summer and winter), however, we think it is necessary to keep both in the main manuscript as we also perform a seasonal comparison. We think that the revised manuscript is now concise enough despite this. Additionally, we have ensured that the main manuscript stands on its own, with clearer summaries and explicit references to the Supplementary Material where needed.

3. I think the models should be referred to as global climate models or Earth System models (ESMs). UKESM is definitely an Earth system model.

We now use consistent terminology across the manuscript with "global climate model".

4. Isokääntä et al., 2022 is regularly referenced. This manuscript appears to use observational data from Isokääntä et al. and extend their analysis techniques to GCMs. In the Introduction the authors could consider including a paragraph clearly summarising of Isokääntä et al. and how this study builds on it.

Isokääntä et al was summarised followingly in the initial manuscript: *"a previous study investigated the in-cloud aqueous phase processing of aerosol in more detail by using relative humidity as a proxy to estimate the cloudiness along the air masses (Isokääntä et al., 2022). This study observed a significant increase in sulfate mass in air masses that had recently been in non-precipitating clouds compared to air masses that had not experienced wet processing during the last 24 hours. Isokääntä et al. (2022) didn't observe, however, significant aqueous phase production of organic mass, likely due to the environment studied (boreal forest), in which production of organics from biogenic sources via gas-phase chemistry is dominating. This is in line with earlier observations made for boreal region in central Sweden (Graham et al., 2020)."*

As this summary did not include statements regarding the wet removal, we have now adjusted it into following, also making sure the main conclusions are summarized: *"A previous study by Isokääntä et al. (2022) used relative humidity (>94 %) as a proxy for in-cloud exposure in boreal air masses and*

*found a pronounced increase in sulfate mass in air masses recently influenced by non-precipitating clouds, while no significant aqueous-phase production of organic aerosol was observed—likely due to dominant gas-phase biogenic sources. This is consistent with findings from central Sweden (Graham et al., 2020). These earlier results suggest that sulfate may be more strongly affected by cloud processing and wet removal than organic aerosol, with removal efficiency likely influenced by factors such as precipitation timing, aerosol type, and the stage of the air mass trajectory. Our study builds on this by exploring these aspects across multiple models and observations, employing the GCM Lagrangian evaluation framework presented by Kim et al (2020)."*

**Technical comments**

General technical comments

- Colloacation => co-location

All instances of this word have now been changed in the main manuscript and supplementary material.

- Airmasses => air masses

As above, all instances have been adjusted.

- Use of 'hexagon'. Is this analogous to a model grid box?

We assume the referee refers to the map figures, like Figure 7. These hexagons refer to the area to which the crossing trajectories have been averaged spatially. Instead of the typical rectangular grid, we used hexagonal grid which is better at keeping the area similar in different parts of the globe (i.e., considering the curvature). Thus, it is indeed analogous to model gridbox. We have now also clarified this in the figure captions by changing "for each hexagon" to "for each hexagonal gridbox". We apply hexagonal gridding to obtain a uniform spatial representation of trajectory endpoints, which removes the need for additional area-weighting. The method is implemented via the *hexbin* function provided in the Python library **cartopy** (Elson et al., 2022), which builds on **matplotlib** (Caswell et al., 2023) for map visualisation.

- Please consider using "particles cm$^{-3}$" instead of "# cm$^{-3}$".

We thank the referee for the suggestion. We have now changed this as suggested.

- For clarity please consider using "updraft velocity" rather than "updraft".

All appropriate instances have been now changed.

Abstract

- This needs to include the main study objectives and the main conclusions, i.e. what does the study tell us about how precipitation impacts aerosol-cloud interactions in GCMs?

Indeed, now with the condensed research questions and revised manuscript overall, the abstract should also reflect these changes. We have now rewritten the abstract into following to better reflect the study aims and main conclusions:

*Global climate models (GCMs) face uncertainties in estimating Earth's radiative budget due to aerosol-cloud interactions (ACI). Accurate particle number size distributions (PNSDs) are crucial for improving ACI representation, requiring precise modelling of aerosol sources and sinks. Using a Lagrangian trajectory framework, we examine how clouds and precipitation influence aerosols during transport, and thereby influence aerosol–cloud relationships in the boreal forest. Two GCMs, the United Kingdom Earth System Model (UKESM1) and ECHAM6.3-HAM2.3-MOZ1.0 with the*

*SALSA2.0 aerosol module (ECHAM-SALSA), are complemented with model-derived trajectories and evaluated against in-situ observations, which are accompanied by reanalysis trajectories. Overall aerosol–precipitation trends are similar between GCMs and observations. However, seasonal differences emerge: in summer, UKESM1 exhibits more efficient aerosol removal via precipitation than ECHAM-SALSA and observations, whereas in winter, the opposite is observed. These differences coincide with key variables controlling aerosol activation, such as sub-grid scale updraught velocities and PNSDs. For example, in winter, removal of total aerosol mass in ECHAM-SALSA was stronger than in UKESM1, coinciding with higher activated fractions and larger sub-grid scale updraught velocities in ECHAM-SALSA. For both GCMs, cloud processing along trajectories increased $SO_4$ mass, mainly in the accumulation mode, consistent with observations and model parametrizations. Discrepancies arise more from differences in PNSDs and updraught velocities than from wet removal parametrizations, an example being the underrepresentation of small particles in UKESM1. While our findings are representative of boreal region with predominantly stratiform precipitation, further work is needed to evaluate their applicability to other regions.*

Introduction

- L53: **and therefore** partly masking….

- L67: "Such frameworks facilitate  the development…" OR "Such frameworks **pave** the way for the development…."

- L85-117: Paragraph 3 is very long and would benefit from being broken up.

These three suggestions from above have now been employed.

- L126-129: *"In this work, the effects of wet processing (wet removal and aqueous phase processing) along air mass trajectories on modelled aerosol size distributions are compared with long-term observations of aerosol size distributions in Hyytiälä, Finland. The observations are combined with ERA-Interim reanalysis trajectories, and the trajectories to be utilized with the GCM variables are calculated with the GCM simulation (nudged to ERA-Interim reanalysis) output meteorology"*

These sentences are really important because they describe the research, but the second sentence is difficult to understand.

We have now tried clarifying our message by adjusting these sentences to "*This study compares the effects of wet processing (wet removal and aqueous-phase processing) on modelled aerosol size distributions with long-term observations from Hyytiälä, Finland. Observational trajectories are based on ERA-Interim reanalysis, while model trajectories are calculated using meteorology data from GCM AMIP-style simulations in which wind fields were nudged to ERA-Interim reanalysis.*".

The placement of this sentence is now also changed in the revised and reorganized introduction.

- L98-99: "*…more efficient removal on the number concentration of the accumulation mode particles with increasing accumulated precipitation…*"

Suggest: "…more efficient removal of the accumulation mode particles with increasing accumulated precipitation…"

This suggestion has been implemented.

- L100: environment => environments

This has been now changed.

Methods:

- L168-169: *"The dataset used in this study is reduced compared to Isokääntä et al. (2022) and extends to the end of 2018 to facilitate comparisons to the simulation period of the GCMs."*

Do you mean a shorter time series? Or have other constraints been applied to the data for quality control purposes? What is the time period used for this study – is it 2005-2018?

Yes indeed, in Isokääntä et al, data until 2019 was used so we employ here a slightly shorter dataset as the GCM simulations only extended to the end of 2018. The referred sentence is now changed to clarify this: *The dataset for particle number size measurements spans 2005–2018, slightly shorter than in Isokääntä et al. (2022), to match the GCM simulation period. The ASCM data extends from 2012 to 2018."*

- L174-178: This sentence needs rewriting for clarity. I suggest describing one model component per one sentence.

This sentence is now rewritten for clarity and each model component is mentioned in separate sentence as suggested.

- L181-185: Please edit this sentence for clarity. I suggest using shorter sentences.

We have now split the sentence into two and shortened it overall for clarity.

- L207-208: "…chemical components including mass mixing ratios of sulfate noted here as SO4 (extracted as sulfuric acid H2SO$_4$ and then converted, see Sect. S1.1),…" See also Section 2.4.3.

  Could the authors please explain why they have taken this approach? UKESM does output mass mixing ratios of sulphate aerosol (SO $^{2-}$) for the different aerosol modes (nucleation, Aitken etc) and I'm not sure why these were not used. UKESM's $H_2SO_4$ diagnostic is sulphuric acid in the gas phase.

We thank the reviewer for pointing this out. In UKESM1, sulfate aerosol ($SO_4^{2-}$) is represented in the output as sulfuric acid ($H_2SO_4$). To make this consistent with the observational quantities, we converted $H_2SO_4$ to $SO_4$ using the molecular weight ratio as described in the supplementary material (S1.1). This ensures that the diagnostic is directly comparable to the instrumentally derived sulfate mass.

We acknowledge that UKESM can provide mass mixing ratios of sulfate aerosol across the different modes (nucleation, Aitken, accumulation, etc.). However, for the purposes of this study, the mass in the Aitken and accumulation modes already provides a good approximation to the instrument response, which is designed to capture the fine-mode sulfate peak. Applying more detailed mapping of the instrument size range onto the multi-modal distribution would require a number of additional assumptions and mathematical operations, but in practice would not significantly improve the comparability.

Therefore, we consider the molar mass conversion to be the most straightforward and robust approach to harmonize model output with the measurements, while remaining faithful to the underlying aerosol representation in UKESM.

- Section 2.2: Could the authors please include a mention of how were the GCM simulations extended beyond 2014 and up to 2018? Was a future simulation used from one of the SSP scenarios used? Although there is little expected difference in SSPs for the 2014-2018 period, it would be useful to see what was used.

The simulations were not extended per se, they were merely continued longer to also cover years

between 2014 and 2018. We have now reworded the text in the manuscript to make this clearer by changing the word "extended" to "longer" in all appropriate instances. Thus, no future simulations were performed.

- L241 By "4 d" do you mean four day?

Yes exactly. We have now clarified this by writing this out as "4-day"

- L277: "...allows to retain the…" => "...permits retention of the…"

Please see our reply below on regarding how we adjusted the whole sentence.

- L277-279: While I understand the reasons for the author's approach here, I'm not sure 'ensuring consistency' is the right term because the precip is ERA, which the aerosol variables are GCM.

Yes, the referee is correct in the sense that for the observations, precipitation is ERA. For GCMs, both aerosols and precipitation originate from the GCM output diagnostics, thus it is not directly apples-to-apples comparison. However, with consistency here we referred to the numerical accuracy only. We have now clarified this by changing the sentence to "*This approach was chosen to retain the original numerical precision of ERA-Interim (and GCM) precipitation data, ensuring accurate alignment with co-located GCM variables (e.g., aerosol size distributions and chemical composition), which HYSPLIT does not provide. Here, "consistency" refers to numerical accuracy rather than matching data sources.*".

- Section 2.4.1: Were the hourly observations averaged to co-locate with the 3 hourly model data?

We did not average the 1-hourly observations, but instead only selected the time points that matched with the GCM data. This was done as the GCM values were also outputted as "instantaneous" values every third hour.

- L300: Please define DMPS here
- L300: Please define ACSM here

Definitions for the two above have now been added.

Results sections

- In general, at the start of a sub-section or paragraph, please try to give an overview of what the relevant figure is showing. The sub-figures can then be referenced as each result is described. Please see the Section 3.1 comments for a suggestion.

We thank for the referee from this feedback. We have now carefully revised the whole result section from the manuscript to accommodate the feedback from the referee. We have also aimed on condensing the text. To accommodate these changes, so text pieces have also been reordered to create more clear, easier-to-follow storyline.

Section3:

- In paragraph 1 consider indicating which variables are be compared between the model and observations - Aerosol particle number size distributions and chemical compositiondd

This has been added to the first paragraph of Section 3.

- L340-342: "For the GCMs, fewer studies looking into aerosol properties at single sites exist, but Leinonen et al. (2022), for example, conducted an extensive study comparing long-term aerosol

particle seasonality and trends in observations and GCMs in multiple locations, also including SMEAR II."

This could be in the Introduction so that the Results sections can focus on the study's analyses

We thank the referee from the suggestion, and indeed the referred sentence is more of an introductory material. However, to accommodate also other referees' suggestions for shortening the text, we have now removed the sentence completely.

- L347-354: I suggest that this could be a short subsection. Something like "Comparison of transport for ERA and GCMs". Figure S4 could be included in the main manuscript with the GCM output shown as difference plots compared with the ERA data. This would also help illustrate the study location.

Section3.1

- L356-363: I suggest re-writing the first paragraph as follows to help guide the reader. Sub-figures can be referenced as the results are described. Information about the time periods plotted and which percentiles are used for the statistics can be clearly stated in the Figure caption.

    In Figure 1 particle number size distributions from the GCMs is compared with observational data at the SMEARII. Median aerosol number concentrations for nucleation, Aitken and accumulation mode are shown in Table S5. ECHAM360 SALSA data in Figure 1c is presented in its native resolution for size bins falling between dp = 3.0 – 1700 nm and those size bins are positioned within the y-axis to the geometric mean of the ECHAM-SALSA size bins (see Table S3). To calculate the difference in Figure 1e, the measured size distribution is regridded to the ECHAM-SALSA bins by integrating between the upper and lower limit of each ECHAM-SALSA size bin.

We thank the referee for these suggestions. To accommodate also the comments below, we have now adjusted Figure 1 by removing the GCM size distributions and only show the differences instead. The first paragraph is now starting as the referee suggest, however, due to this figure adjustments some technical details are removed as those became unnecessary. In addition, to accommodate the other referees regarding shortening of the text, we have now also removed repetitive technical details from the text if those were already included in the figure captions.

- L375: *"…this could indicate from other processes which dominate the accumulation mode."*

I suggest re-writing as: "…this could indicate that there are other processes that dominate the accumulation mode."

This has been rewritten as suggested.

- Figure 1: Consider reducing the number of sub-figures. For example, omit 1b and 1c and just show the difference plots (1d and 1e).

We have now removed the subfigures 1b and 1c and show the differences instead.

Section 4

- I suggest that Section 4 is re-written to help guide the reader through it and to help the reader understand the main results. I suggest that the analysis of the impact of extracting the vertically resolved liquid precipitation from UKESM1 described in paragraphs 1 and 2 is described in its own section, e.g. 4.1. I then think that including the analysis of the representativeness of precipitation in UKESM and ECHAM-SALSA compared to the AeroCom GCMTraj ensemble here

is distracting. I think this analysis, together with Appendix B, could be moved to the supplementary material.

We are very grateful for the referee from these detailed suggestions to improve our manuscript. We have now moved the Appendix B and the text from the beginning of Sect. 4 concerning the other AeroCom models into the supplementary material (new section S4). To accommodate this, we also added a sentence in the conclusions: "Supplementary analysis comparing a wider ensemble of GCMs indicated that these two models were broadly representative, with their aerosol–precipitation relationships generally falling near the middle of the inter-model spread."

Text discussing Appendix A (vertically resolved precipitation) has been made into its own section as suggested.

- L440: "smallest" => smaller

This has been modified. Note that this text locates in the revised supplementary material.

- L445-451: I suggest removing the sentence "*The analysis is simplified by removing the size dependent component noted in previous literature (see e.g., Figure 3 in Isokääntä et al., 2022 and Figure 4 in Khadir et al., 2023)…*"

I don't understand what the sentence is referring to here, but it could be included when the relevant results are shown later.

We have removed this sentence.

- L445-451: Based on the above comments I then suggest starting Section 4 with the following paragraph:

"In this section we use a Lagrangian trajectory framework to…. In Section 4.1 we examine the impact of using vertically resolved liquid precipitation, which has not previously been done for Lagrangian trajectory analyses. Then the investigation inspects the relationship between accumulated precipitation and aerosols for the two GCMs used in this study: UKESM1 and ECHAM- SALSA. In Section 4.2 we focus on total aerosol mass and number. In Section 4.3 we focus on the OA, BC, and SO4 portions of the total mass for submicron-size aerosols. Then, in Section 4.4, the processes controlling the precipitation-aerosol relationships are investigated, and the differences are discussed in detail between the GCMs and within each GCM."

This section has now been reorganized and starts with the description the referee suggested.

- Section4.1: To help the reader understand the results in this section I think it would be helpful to describe them after Figure 4. Why you have looked at seasonal periods (as in L466-474) can then be discussed as you discuss the results shown in Figure4. Additional results such as those in Figures S7-9 can be mentioned at the end of the Section. I also think that statements like *"Non-normalized mass and number concentrations are shown in Figures S8 and S9, and those are similar to Isokääntä et al., (2022) which employed whole year data and total precipitation"* (L475-476) are not useful. How are the mass and number concentrations similar to Isokääntä et al., (2022) and why is this important? The novelty of this study is the model-observation comparison. If it is important the observational data used here is similar to Isokääntä et al., (2022), there should be an additional analysis.

We placed all figures after the text they are first mentioned to be able to use the automatic referencing in word. We understand this is sometimes inconvenient if one has to go down a lot of text before seeing the figure. However, we have now aimed on shortening the text and applied the restructuring suggestions given by the referee. In addition, we have removed the unnecessary technical descriptions (repetitive of the figure captions) from the text (thus also reflecting the other reviewers' wishes regarding conciseness of the work) and start the section by describing the

results.

Indeed, the comparison to Isokääntä et al is not in focus of this work, and we used it mainly as a double check to make sure our results (for observations) are not biased due to the reduced dataset when compared to Isokääntä et al. However, as the referee mentions, these analyses are not key information. We have therefore reduced the sentence to "Non-normalized mass and number concentrations are shown in Figures S7 and S8". (note changed figure numbering).

- L471-474:

  (i)    "…which further cause differences..." => "…which causes further differences..."

  (ii)    I suggest splitting this sentence up for readability.

We appreciate the referee for noting us as the sentence was indeed unnecessary long. We have rewritten it into *"Normalization of the median mass/number concentration to the median value under zero accumulated stratiform precipitation is used in this study. This approach aims to minimize the influence of differences in the native particle number size distributions (e.g., Figure 1), which affect total mass and number concentrations, and instead highlight the removal attributable to precipitation."*

- L477: In describing Figure 4 here and elsewhere in the manuscript the authors refer to "removal of the normalized masses…" However, the plots are not actually showing removal processes. I think it would be more correct to refer to e.g. reductions in normalized masses.

We thank the referee for noting this, as in many places we have tried to write about "aerosol-precipitation-relationships" to avoid exactly this misleading terminology. We have now carefully revised the text and fixed in instances where "removal" was used, we have replaced the word with "reduction" whenever appropriate (i.e., we still use "wet removal" and similar when discussing the processes in general)

- L480-483: Why would there be a seasonal effect of UKESM's missing boundary layer nucleation process?

In the lines referee mentions, we referred to the fact that in summer, NPF (in general) contributes a lot to the concentration of the smallest particles. Now, with UKESM1 missing this component by missing boundary layer nucleation, summer and winter behave in similar manner. We have rewritten the sentence for clarity: "ECHAM-SALSA and the observations show clear seasonal differences in particle number removal, with much more efficient removal in winter than in summer. UKESM1, however, does not display this seasonal contrast—likely because it lacks boundary layer nucleation, a key source of small particles during summer."

- L539 and Figure 5 caption: "…number of data rows for the bin was 10 or larger." What does this mean?

For the bins which have high accumulated precipitation, the number of data rows (i.e. trajectories) is more limited. To ensure we have enough datapoints for each bin, we applied a limit of 10 trajectories per bin as a minimum. This is what the sample size subplot shows, i.e., the tiniest bar visible has 10 observations (=trajectories) in it. We have now reworded the "data rows" into "trajectories" to clarify this.

- L575: Activated fraction? Is this the fraction of $N_{80}$ particles that have been transported upward?

The activated fraction here (referring to Fig. 7k-o) is defined as the fraction between the number of activated particles (Nact, as obtained from the GCMs and co-located to the trajectories) to the total number of particles (Ntot, as obtained from the GCMs and co-located to the trajectories) at the height of the trajectory. N80 is calculated from the full PNSD (also co-located to the trajectories) and

is shown separately to visualize the number of particles that could potentially activate, thus the dp>80nm limit selection. We have also clarified the sentence: "*The activated fraction (Nact/Ntot)..*"

- L609-610 "These regions coincide with the high trajectory frequencies; thus, the high updraughts are being reflected on the averages irrespective of transport direction in Figure 7j."

Could the authors please clarify this statement.

We have now rewritten the sentence to "*These regions coincide with areas of high trajectory frequency, meaning most air masses pass through them. As a result, the elevated updraught velocities in these regions strongly influence the averages shown in Figure 7j.*"

- L614: "…where the north of Norway…" The region of very high activated fraction appears to extend over the Arctic Ocean.

Yes indeed, we thank the referee for noting this. We now also mention the arctic ocean in this context in the rewritten sentence.

- L621-623: "*Assuming the solubility of OA compensates for the missing portion of SO4 in summer, these differences could increase the particle activation potential even further during summer in UKESM1 (compared to winter).*"

I think this is a bit difficult to follow.

The sentence has now been reworded into "*If the higher solubility of OA in summer compensates for the lower SO$_4$ levels, this could further enhance the particle activation potential in UKESM1 during summer compared to winter.*"

- L624-626: Please consider re-wording as "Figure 8 shows the relationship between mean activated fraction and mean updraft velocity that the air mass experienced before arriving at SMEAR II for the summer and winter periods."

The sentence is now reworded as suggested.

- Figure 8: Please consider resizing the text annotations on the plots.

We have now adjusted both figure and font sizes in all of the manuscript figures for clarity, also considering the feedback from other referees.

- L634: Here, I think it should be "lower concentrations" rather than "smaller concentrations"

Indeed, we thank the referee for noting this. It has now been corrected.

- L649: "…which is steeply…" => "…which it steeply…"

This is also now corrected.

- L650-651: "*As these differences are taking place relatively close to SMEAR II, it is likely that they are driven by the seasonal differences in the transport very close to SMEAR II.*"

Could the authors please explain why that might be?

We thank the reviewer for this question. The seasonal differences in updraughts near SMEAR II likely reflect a combination of local conditions and transport characteristics in ECHAM-SALSA. In JJA, the increase around 18 hours before arrival is associated with higher updraughts over Norway and Sweden along many trajectories, while lower updraughts over the nearby ocean explain the decrease between 18 and 12 hours; updraughts then strengthen again closer to SMEAR II. In DJF, stronger updraughts over the ocean and SMEAR II's proximity to these marine regions likely contribute to the increase from 12 to 0 hours, with the 36-hour peak reflecting trajectories crossing ocean regions with elevated updraughts. Seasonal land–ocean contrasts may arise from stronger surface warming and boundary-layer convection in summer, and from low-level clouds and turbulence induced by oceanic temperature inversions in winter. Local

boundary-layer dynamics and modest orographic effects near SMEAR II could further modulate updraughts. While some aspects remain speculative, these factors together offer a plausible account of the observed seasonal patterns.

We have adjusted the sentence to "*As these differences are taking place relatively close to SMEAR II, it is likely that they are driven by the seasonal differences in the transport and local conditions very close to SMEAR II.*"

- L722: I suggest re-wording "*…UKESM1 updraughts are below and above 0.2 m s-1…*" to "*…UKESM updraught velocities range from 0.2-0.4 m s$^{-1}$…*"

This has been reworded as suggested.

- L749-768: Please consider putting this text in a new sub-section with a heading something like "Other reasons for model differences"

We thank the referee for this suggestion, and we have now made this section as its own subsection.

- L826-829: Possibly, but this would be balanced against likely decreases in SO2 emissions due emission control policies in many regions.

Indeed, it is not straightforward what might happen. We have adjusted the sentence into following: "*Recent findings from the Holuhraun eruption (Jordan et al., 2023) also suggest aqueous-phase oxidation dominates SO$_2$-to-SO$_4$ conversion in GCMs. While future increases volcanic activity (Chim et al., 2023), could enhance SO$_2$ levels and boost in-cloud SO$_4$ production, ongoing emission controls may reduce anthropogenic SO$_2$, potentially counteracting this effect and influencing aerosol size and composition*"

- L837-846: Please consider putting this paragraph before the preceding discussion on SOA

The placement of these paragraphs is now changed as suggested.

- L835-834: In this study region or in general?

In general. We refer here to isoprene dominated environments (our work is focused on monoterpene dominated forest). We have reworded the sentence into "*For isoprene-dominated environments, the formation of aqSOA is a significant source for total SOA burden (e.g., Lamkaddam et al., 2021).*"

- L844-846: Are the same (e.g. CMIP6) emissions data sets being used in both models? Would rule out differences in emissions.

Yes indeed, the same CMIP6 emission data sets are used in both models that being also the requirement in the AeroCom Phase III GCM Trajectory Experiment, thus ruling out emissions differences. We added a sentence noting this to the text: "*Both GCMs employ CMIP6 emission datasets as noted in the model setup for AeroCom Phase III GCM Trajectory Experiment, and thus the differences observed here unlikely arise from varying emissions.*"

- Figure 11: I think that it would be useful to include the observations for the for corresponding period and sector.

We thank the referee for the suggestion. Initially, we wanted to exclude the observations from this figure as they are essentially presented in Isokääntä et al (and we had them included in the supplementary material). However, we do understand that showing them side-by-side is useful for easy comparison. Therefore, we have now added the observations to Figure 11, and adjusted the supplementary figures accordingly.

- L896-904: Please consider re-wording this paragraph because I cannot understand what it means.

We apologize our initial wording was unclear. We have now reworded the paragraph into following: *"The observed changes in particle number size distributions (Figure 11) reflect the actual model parameterizations. In UKESM1, $SO_4$ produced via aqueous-phase chemistry is allocated to the soluble accumulation mode (dp > 100 nm) and coarse mode (dp > 500 nm) (Mann et al., 2010), with the results here showing increases in the 100–600 nm range. In ECHAM-SALSA, aqueous-phase $SO_4$ is distributed across soluble size bins spanning 50–10000 nm (2a bins; see Table S3, Bergman et al., 2012), with sector-dependent mass increases observed between 50–1700 nm. . In terms of aqueous-phase oxidation of $SO_2$, both GCMs have similar parametrizations, and for example, oxidation of $SO_2$ by ozone ($O3$) and hydrogen peroxide ($H_2O2$) is considered in both (Bergman et al., 2012; Hardacre et al., 2021)."*

- L902-904: Both models have very limited aqueous phases schemes – do the results imply that this does not matter too much, at least at the Boreal forest location?

This is a good question, and unfortunately, we cannot provide a solid answer. Based on our results, we could say that the current parametrizations related to aqueous-phase sulfate production seem sufficient—at least when considering how we observe it in our framework. However, due to our study region, our results are not conclusive regarding the aqueous-phase SOA for two reasons; 1) SOA representation was simplified by parameterisations in the models i.e., it was not explicitly modeled, and 2) our study region is in the monoterpene-dominated area, which has not been shown to exhibit significant formation of aqSOA (opposite has been recorded for isoprene-dominated environments).

Section 5

- In common with Section 4, there are many references to the supplementary materials. I think the text could generally be improved by clearly summarizing the main findings as shown in the main manuscript's figures.

We have now aimed to revise Section 5 in similar manner as sections 3 and 4. In summary, we adjusted the text such that main results are mentioned first with figure reference, followed by more detailed descriptions. Text has been also adjusted overall to make it shorter to accommodate other referees' feedback. Please also have a look on few of the comments above, as they were essentially for Sect. 5.

Conclusions and outlook

- I suggest that this section is edited to ensure that the main results and their impact is clear to the reader.

Indeed, now with the revised introduction and results, the conclusion section also needed adjustments. We have also tried to shorten the text to accompany other referees' comments. Please note that we have now also separated the conclusions and outlook into separate sections to accommodate the other referee.

- I also think that the authors should include some discussion on the impact of their results for GCMs. For example:

  o What were the main model biases against observations?

  o Are there any processes that should targeted for model development?

  o Are the results likely to be applicable to different locations? This is difficult, but

important because GCMs are used for global climate model prediction.

We thank the referee for raising these relevant points. Based on our results, the largest discrepancies are due to the representation of the PNSD. Therefore, we have added a sentence to the conclusions stating this explicitly: *"A primary model bias identified in this study is the difference in aerosol number size distributions compared to observations, particularly the underrepresentation of small particles in UKESM1."*

This follows up to the next bullet point, as it is clear that the aerosol representation in GCMs still need more work, and further comparisons to measurements, in different environments, are essential. This is mentioned in the revised conclusions: *"This emphasizes the need for better representation of particle number size distributions (PNSDs) in GCMs."*

The location dependence is indeed very important, and it was partly mentioned in the discussion when describing the type of precipitation (stratiform vs convective). We have now added a sentence to the conclusions mentioning this explicitly: *"In such environments, the vertical distribution, intensity, and frequency of precipitation could differ substantially, potentially altering the accumulated wet removal along trajectories. Therefore, while our results are representative of boreal regions with stratiform precipitation, further work is needed to assess how applicable they are to regions with different precipitation regimes."*

Supplementary

- Tables S1 and S3: I'm missing a description of "GSD" and "GMD" in column 4.

We thank the referee for noting these. Descriptions for both have now been added (GSD = geometric standard deviation and GMD = geometric mean diameter)

- Figures S4 and S5: Difference plots would help to illustrate the differences, or lack thereof, between ERA and the model simulations.

Indeed, the referee is correct difference plots would be helpful here. We have now adjusted both figures as suggested. Please note that Fig. S4 is now presented in the main manuscript as the referee suggested earlier. Total number of figures stayed the same as we moved the figure comparing all AeroCom model to the supplementary material.

**References**

Caswell, T.A., Lee, A., Droettboom, M., Andrade, E.S. de, Hoffmann, T., Klymak, J., Hunter, J., Firing, E., Stansby, D., Varoquaux, N., Nielsen, J.H., Root, B., May, R., Elson, P., Seppänen, J.K., Dale, D., Lee, J.-J., Gustafsson, O., McDougall, D., hannah, Straw, A., Hobson, P., Lucas, G., Gohlke, C., Vincent, A.F., Yu, T.S., Ma, E., Silvester, S., Moad, C., Kniazev, N., 2022. matplotlib/matplotlib: REL: v3.6.2. https://doi.org/10.5281/zenodo.7275322.

Elson, P., de Andrade, E. S., Lucas, G., May, R., Hattersley, R., Campbell, E., et al. (2022). Scitools/cartopy: V0.20.3 [Software]. Zenodo. https://doi.org/10.5281/zenodo.6775197

---

## Author Response (AR2)

**Referee feedback in black, author responses in blue**

The authors have made extensive revisions to the manuscript, which is now much better organised and easier to follow. This is to be commended. However, I do think that the conclusions section needs to be strengthened. I have made General comments below regarding the Conclusions in Section 6, as well as some minor Technical comments, and once these have been addressed I recommend that the manuscript is published.

We thank the referee for thoughtful feedback and positive assessment of the revisions. We appreciate your constructive suggestions regarding the Conclusions section and the technical comments, and have now addressed these points carefully in this revision.

In addition to the requested changes, two technical corrections were applied,

- 1. the author affiliations were corrected
- 2. explanation for CCN (cloud condensation nuclei) was added when first mentioned (in page 20)

**General comments:**

Conclusions regarding Objective 1.

I think the sentences in e.g. L843-844, L853-854 do not summarise the study's findings clearly.

From my interpretation of Figure 4 the models do predict the observed trajectory-based relationships between particle mass and precipitation for total aerosol and the three aerosol species. However, there are model biases in trajectory-based relationships between particle number and precipitation that are different for the model and season.

So I think that the Conclusion should state that: UKESM exhibited significant loss in particle number via precipitation compared to the observations (and ECHAM) in summer. The lack of boundary layer nucleation (BLN) likely contributes to more, large particles that rain out more easily. Figures 7 and 8 also show that the strong relationship between activated fraction and updraft velocity in UKESM in summer may also contribute to the model's bias in number concentration.

We do agree with the referee regarding this conclusion, and apologize that our writing was not yet appropriate enough such that the message would be clear. Please see our reply at the end of these general comments on how the referred text was revised.

However, in winter both models overpredict particle number concentration with accumulated precipitation compared to the observations. The lack of BLN may contribute to UKESM's bias, but I'm curious if both model's overprediction of number concentration could be due to underprediction of accumulated precipitation <2mm - where the largest model biases in number concentration occur (Fig 4c)?

We thank the referee for this very careful observation regarding the differences in number of trajectories with certain number of trajectories. Indeed, the overprediction during winter could be partly due to the difference in the smaller number of trajectories with accumulated precipitation below 2mm (as can be seen from Fig. 4c where purple bars i.e., observations, have larger count compared to the models). This could cause the fact that the particle number removal is not as efficient in the models as in the observations during wintertime. On the other hand, with larger accumulated precipitations the models show larger trajectory counts, which in turn can contribute to the fact that the normalized number concentrations approach each other's (Fig. 4b) with increasing accumulated precipitation. Therefore, the differences in the precipitation are unlikely the only explanation,

however, this, indeed, could contribute to the differences. We have now added a sentence to results section 4.4.3 acknowledging this (starting from line 632): "However, some of the winter differences may also be attributed to variations in the number of trajectories with specific amounts of accumulated precipitation (Fig. 4c). Observations show a higher frequency of trajectories with low accumulated precipitation (<2 mm), whereas the models produce slightly more trajectories with larger precipitation totals."

This is now also brought up in the revised conclusions which were also adjusted as suggested by the referee. For this, please see our reply below.

The role of activated fraction, updraft velocity, and the relationship between them (Figures 7 and 8), on model biases in trajectory-based relationships between particle number and accumulated precipitation in winter are less clear to me. For example, I don't follow that: "The seasonal differences we observed in these variables, along with changes in particle chemistry during the transport, were consistent with the aerosol-precipitation relationships." (L853-854). Are the model biases due to too much large aerosol, or a too-weak relationship between activated fraction and updraft velocity – perhaps because the GCM's do not resolve the local meteorology well?

The model biases arise from both effects (aerosol number & relationships between activated fraction and updraught velocity) the referee mentions. With the current set up in our work, however, it is not possible to perfectly distinguish the actual roles of these factors such that we would b able to determine which has the largest effect—these are also impacted by each others.

Regarding the referred lines in the previous version of the manuscript (L843-844, L853-854) we have now revised the second and third paragraph of the conclusions completely and arranged them into three paragraphs for clarity. We also added a reference to a very recent study by Virtanen et al., 2025, which highlights the inter-model differences (and differences to observations) in the relationships between droplet number and updraughts.

"Our first objective was to investigate whether trajectory-based relationships between aerosol mass, number, and precipitation differ between observations and GCMs. For aerosol mass, the observed removals generally fell between those simulated by ECHAM-SALSA and UKESM1 across seasons, indicating that both models captured the observed mass—precipitation relationship for total aerosol and individual species (OA, SO4, BC). In contrast, aerosol number revealed clear model biases that varied by season. In summer, UKESM1 exhibited a pronounced loss of particle number via precipitation compared to both observations and ECHAM-SALSA. This bias likely stems from the absence of boundary layer nucleation, which produces fewer small particles and leaves a larger fraction of particles susceptible to wet removal.

Key variables influencing the wet removal processes, such as number of potential cloud condensation nuclei ( $N_{80}$ ) and updraught velocities, were also examined to evaluate the observed removals. In UKESM1, a strong summer correlation between activated fraction and updraught velocity (Figs. 8) may further increase particle number removal. However, analogous study examining droplet number/CCN versus updraught (Virtanen et al., 2025) show substantial variability across models, highlighting that the relationship between updraught and particle activation remains uncertain and warrants further investigation. In winter, both models overpredicted particle number removal relative to observations. This overprediction may in part reflect differences in precipitation statistics, with models simulating fewer low-precipitation trajectories (<2 mm) than observed (Fig. 4c). However, other factors such as particle size distributions, activation efficiencies, and limitations in the representation of subgrid-scale

meteorology are also likely to contribute. Overall, our results emphasize the need for better representation of particle number size distributions (PNSDs) in GCMs.

Earlier work has indicated that aerosol activation into cloud droplets followed by rainout is the dominant wet removal process. Our results support this, with UKESM1 showing nucleation followed by rainout as the largest contributor. Supplementary analysis comparing a wider ensemble of GCMs indicated that these two models were broadly representative, with their aerosol—precipitation relationships generally falling near the middle of the inter-model spread. Overall, our method using normalized submicron mass and number as a function of accumulated precipitation proved to be effective in comparing removal across models, though it lacks details on particle size evolution—an important topic for future work."

**Technical comments:**

**Section 1: Introduction**

**1. L141-142:**

"Do the GCMs exhibit similar increase in sulfate mass due to in-cloud production as the observations and are the observed effects reasonable when reflected to model parametrizations?"

Suggest compared instead of "reflected"

The word "reflected" is now changed to "compared" as suggested.

2. L143:

out -> our

We thank the referee for noting this typo. It has now been corrected.

3. L145:

Please define the SMEAR II acronym here or refer to e.g. 'the measurement station'.

Indeed, this is the first time when the acronym is used. We have now adjusted the sentence to "The aerosol properties at the measurement station (Hyytiälä, Finland) are given...".

**Section 2: Data and Methods**

4. L190:

"...new particle formation in the boundary layer is not yet implemented in UKESM1 (Mulcahy et al., 2020)."

While not a correction as such, boundary layer nucleation is now available in UKESM1.1 and will be released with the next UKESM version.

This is a good point indeed. We adjusted the sentence to reflect this "..new particle formation in the boundary layer is not implemented in this version of UKESM1 (Mulcahy et al., 2020)"

5. L192-193:

"...were ran longer to cover years from 2005 to 2018..."

I suggest: "...were extended for the period 2005 to 2018..."

The given suggestion is now implemented.

**6. L242:**

**GCMoutput -> GCM output**

We thank the referee for noting this typo. The missing space is now added.

**Section 4**

**7. L438:**

"...thus unlikely driving differences..." -> "...thus unlikely to be driving differences..."

This has been adjusted as suggested.

**8. L509:**

Figure 4 and 4.-> Figure 4 and 5.

We thank the referee for noting this typo, it has now been corrected.

**9. L850:**

"Aerosol activation into cloud droplets followed by rainout appears to be the dominant removal process"

Dominant wet removal process

The word "wet" has now been added to the sentence as suggested.

**References:**

Virtanen, A., Joutsensaari, J., Kokkola, H., Partridge, D. G., Blichner, S., Seland, Ø., Holopainen, E., Tovazzi, E., Lipponen, A., Mikkonen, S., Leskinen, A., Hyvärinen, A.-P., Zieger, P., Krejci, R., Ekman, A. M. L., Riipinen, I., Quaas, J., and Romakkaniemi, S.: High sensitivity of cloud formation to aerosol changes, Nat. Geosci., 1–7, https://doi.org/10.1038/s41561-025-01662-y, 2025.